# Lymphotoxin α fine-tunes T cell clonal deletion by regulating thymic entry of antigen-presenting cells

Noëlla Lopes[1], Jonathan Charaix[1], Oriane Cédile[2], Arnauld Sergé[3] & Magali Irla [1]

Medullary thymic epithelial cells (mTEC) purge the T cell repertoire of autoreactive thymocytes. Although dendritic cells (DC) reinforce this process by transporting innocuous peripheral self-antigens, the mechanisms that control their thymic entry remain unclear. Here we show that mTEC-CD4$^+$ thymocyte crosstalk regulates the thymus homing of SHPS-1$^+$ conventional DCs (cDC), plasmacytoid DCs (pDC) and macrophages. This homing process is controlled by lymphotoxin α (LTα), which negatively regulates CCL2, CCL8 and CCL12 chemokines in mTECs. Consequently, *Ltα*-deficient mice have increased expression of these chemokines that correlates with augmented classical NF-κB subunits and increased thymic recruitment of cDCs, pDCs and macrophages. This enhanced migration depends mainly on the chemokine receptor CCR2, and increases thymic clonal deletion. Altogether, this study identifies a fine-tuning mechanism of T cell repertoire selection and paves the way for therapeutic interventions to treat autoimmune disorders.

[1] Centre d'Immunologie de Marseille-Luminy, INSERM U1104, CNRS UMR7280, Aix-Marseille Université UM2, Marseille 13288 cedex 09, France. [2] Institute of Molecular Medicine, Department of Neurobiology Research, University of Southern Denmark, J.B. Winsløwsvej 25, 5000 Odense C, Denmark. [3] Centre de Recherche en Cancérologie de Marseille, Institut Paoli-Calmettes, INSERM U1068, CNRS UMR7258, Aix-Marseille Université UM105, 13273 cedex 09 Marseille, France. Correspondence and requests for materials should be addressed to M.I. (email: magali.irla@inserm.fr)

Thymic clonal deletion, called negative selection, prevents the generation of autoreactive T cells that could induce autoimmunity. The thymus is subdivided into a medulla surrounded by a cortex, both compartments dedicated to specific selection processes[1,2]. The medulla has a key function in purging the T cell repertoire of self-reactive specificities, through the large diversity of self-antigens (Ag) expressed by medullary thymic epithelial cells (mTEC)[3–6]. The cortex also promotes the deletion of autoreactive T cells[7,8]. It has been estimated that ~75% of negatively selected cells are deleted at the double-positive (DP) stage in the cortex, and that ~25% are deleted at the single-positive (SP) stage in the medulla[7]. Dendritic cells (DC) are involved in this process since their constitutive ablation results in impaired clonal deletion and fatal autoimmunity[9]. DCs are involved in the deletion of both DP thymocytes in the cortex and SP thymocytes in the medulla[10]. They constitute a heterogeneous population comprising three distinct subsets: CD11c$^{int}$BST-2$^{hi}$ plasmacytoid DCs (pDC) and two CD11c$^{hi}$ conventional DC (cDC) subsets—CD11c$^{hi}$CD8α$^{hi}$SHPS-1$^-$ resident and CD11c$^{hi}$CD8α$^{lo}$SHPS-1$^+$ migratory cDCs[11,12]. Intrathymically derived resident cDCs, located in close proximity to mTECs, possess the ability to cross-present self-Ags, expressed by mTECs, to thymocytes[13–15]. Although mTECs express a large array of self-Ags that critically contributes to negative selection, they cannot cover the whole spectrum of self-Ags expressed in peripheral tissues[5,16]. Migratory cDCs and pDCs reinforce the deletion of autoreactive thymocytes by continuously migrating from the blood to the thymus, where they display peripheral self-Ags that would be otherwise not presented to thymocytes[17–21].

While the migration of cDCs and pDCs in the thymus was described to depend on CCR2 and CCR9, respectively[18,19], the implication of the thymic microenvironment and more specifically that of mTEC-thymocyte crosstalk in this process remains unknown. Furthermore, although thymic macrophages constitute another type of Ag-presenting cells (APC) that has long been associated with the clearance of apoptotic thymocytes[22,23], the mechanisms that sustain their thymic entry as well as their respective contribution in clonal deletion remain elusive.

Here, we show that Ag-specific interactions between mTECs and CD4$^+$ thymocytes regulate the thymic entry of peripheral SHPS-1$^+$ cDCs, pDCs and F4/80$^+$CD11b$^+$ macrophages. This phenomenon is tightly controlled by the tumour necrosis factor (TNF) member, lymphotoxin α (LTα), induced in CD4$^+$ thymocytes upon crosstalk, which represses CCL2, CCL8 and CCL12 expression in CD80$^{lo}$ mTECs. We observed that increased expression of these chemokines in CD80$^{lo}$ mTECs from $Lt\alpha^{-/-}$ mice ($Lt\alpha^{-/-}$ mTEC$^{lo}$) correlates with an upregulation of c-Rel and p65 classical nuclear factor-kappa B (NF-κB) subunits, previously described to regulate CCL2 and CCL8[24–26]. Enhanced thymic recruitment of peripheral DCs and macrophages in $Lt\alpha^{-/-}$ mice is drastically reduced in the absence of the chemokine receptor CCR2. We also show that CCR1 and CCR5 control the thymic pool of these cell types, but at a lesser extent than CCR2. Importantly, we demonstrate that LTα-regulated DC and macrophage thymus homing fine tunes the deletion of autoreactive thymocytes in both the cortex and medulla. Finally, we show that migratory cDCs and macrophages are more competent in vivo for deleting autoreactive thymocytes than pDCs, a phenomenon accentuated on a $Lt\alpha$-deficient background. Altogether, this study reveals an unexpected role for mTEC-thymocyte crosstalk in controlling the thymic entry of peripheral DCs and macrophages. This process, tightly regulated by LTα, which in turn represses CCL2, CCL8 and CCL12 expression, substantially impinges clonal deletion.

## Results

**mTEC-CD4$^+$ T cell crosstalk regulates thymic homing of APC.** To investigate whether interactions between mTECs and autoreactive CD4$^+$ thymocytes regulate the intrathymic pool of DCs and macrophages, we used OTII-$Rag2^{-/-}$ mice, which express a MHCII-restricted transgenic TCR specific for ovalbumin (OVA), and RipmOVAxOTII-$Rag2^{-/-}$ mice, carrying a RipmOVA transgene that drives the synthesis of membrane-bound OVA specifically in mTECs. Consequently, OVA is only expressed in mTECs from RipmOVAxOTII-$Rag2^{-/-}$ mice, in which Ag-specific interactions with OTII CD4$^+$ thymocytes can occur[27]. We found an ~1.5–3-fold increase in frequencies and numbers of CD11c$^{hi}$BST-2$^{lo}$ cDCs and CD11c$^{int}$BST-2$^{hi}$ pDCs from RipmOVAxOTII-$Rag2^{-/-}$ compared with OTII-$Rag2^{-/-}$ thymi (Fig. 1a, Supplementary Fig. 1). Among the two cDC subsets, whereas numbers of CD8α$^{hi}$SHPS-1$^-$ resident cDCs were similar in both mice, numbers of CD8α$^{lo}$SHPS-1$^+$ migratory cDCs were higher in RipmOVAxOTII-$Rag2^{-/-}$ than in OTII-$Rag2^{-/-}$ mice (Fig. 1b, Supplementary Fig. 1). We also found increased frequencies and numbers of F4/80$^+$CD11b$^+$ macrophages in these mice (Fig. 1c). Similar results were obtained by transplanting OTII-$Rag2^{-/-}$ bone marrow (BM) cells into lethally irradiated OTII-$Rag2^{-/-}$ and RipmOVA-$Rag2^{-/-}$ recipients (OTII:OTII and OTII:RipmOVA chimeras, respectively) (Supplementary Fig. 2a-d). Three weeks later, an ~2-fold increase in SHPS-1$^+$ cDCs, pDCs and macrophages was observed in the thymus from OTII:RipmOVA compared with OTII:OTII chimeras (Supplementary Fig. 2b–d), confirming that the crosstalk between OVA-expressing stromal cells and CD4$^+$ thymocytes controls the cellularity of these three cell types. This phenomenon was not due to increased proliferation, since similar frequencies of proliferating Ki-67$^+$ SHPS-1$^+$ cDCs, pDCs and macrophages were observed in OTII-$Rag2^{-/-}$ and RipmOVAxOTII-$Rag2^{-/-}$ mice as well as in OTII:OTII and OTII:RipmOVA chimeras (Fig. 1d, Supplementary Fig. 2e).

To test whether increased numbers of these three cell types could result from enhanced thymic entry, CD45.1 nucleated blood cells were adoptively transferred intravenously (i.v.) into sublethally irradiated OTII-$Rag2^{-/-}$ and RipmOVAxOTII-$Rag2^{-/-}$ recipients (Fig. 1e). Three days later, we found an ~2-fold increase in frequencies and numbers of CD45.1 donor cells into RipmOVAxOTII-$Rag2^{-/-}$ compared with OTII-$Rag2^{-/-}$ thymi (Fig. 1f). These donor cells contained increased numbers of SHPS-1$^+$ cDCs, pDCs and macrophages (Fig. 1g, h). Of note, donor cDCs and macrophages expressed higher levels of the CD45.1 marker than pDCs (Supplementary Fig. 3). Similar results were obtained by adoptively transferring CD45.1 splenic DC and macrophage-enriched cells into nonirradiated OTII-$Rag2^{-/-}$ and RipmOVAxOTII-$Rag2^{-/-}$ mice (Supplementary Fig. 4). Consistently, SHPS-1$^+$ cDCs, pDCs and macrophages were substantially reduced in the blood from RipmOVAxOTII-$Rag2^{-/-}$ compared with OTII-$Rag2^{-/-}$ mice ($p < 0.0001$ for SHPS-1$^+$ cDCs and $p < 0.01$ for pDCs and macrophages by unpaired Student's $t$-test, Supplementary Fig. 5). Altogether, these data indicate that the thymic entry of peripheral DCs and macrophages is controlled by mTEC-CD4$^+$ thymocyte crosstalk.

**LTα limits APC thymic entry through CCR2 ligands in mTECs.** We investigated the underlying mechanism(s) by which mTEC-CD4$^+$ thymocyte crosstalk regulates the thymic recruitment of peripheral DCs and macrophages. Since three TNF members, RANKL, CD40L and LTα1β2, are implicated in mTEC-thymocyte crosstalk[12,28–31], we examined whether they could be regulated upon Ag-specific interactions with mTECs. In contrast to *Tnfsf11* (RANKL) and *Cd40lg* (CD40L), we found that only *Lta*

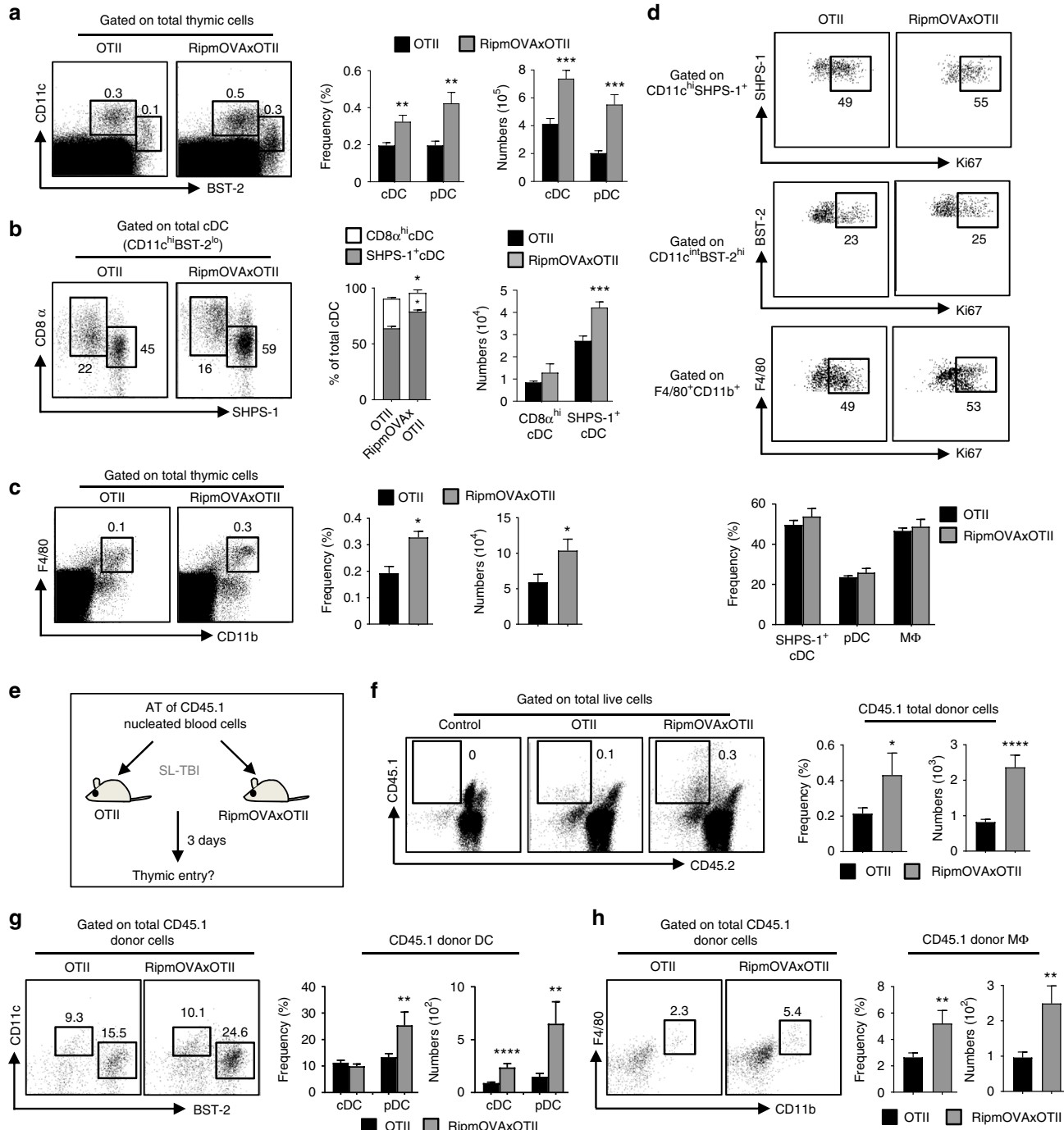

**Fig. 1** Ag-specific interactions between mTECs and CD4[+] T cells increase the thymic entry of circulating DCs and macrophages. **a–c** Flow cytometry profiles, frequencies and numbers of cDCs (CD11c[hi]BST-2[lo]), pDCs (CD11c[int]BST-2[hi]) (**a**), resident cDCs (CD8α[hi]SHPS-1[−]), migratory cDCs (CD8α[lo]SHPS-1[+]) (**b**) and macrophages (F4/80[+]CD11b[+]) (**c**) in the thymus from OTII-*Rag2*[−/−] and RipmOVAxOTII-*Rag2*[−/−] mice. Data are representative of three independent experiments (*n* = 3 mice per group and per experiment). **d** Flow cytometry profiles and frequencies of proliferating Ki-67[+] thymic DC subsets and macrophages. Data are representative of two independent experiments (*n* = 3 mice per group and per experiment). **e** Experimental setup: nucleated blood cells from CD45.1 WT congenic mice were adoptively transferred into sublethally irradiated CD45.2 OTII-*Rag2*[−/−] and RipmOVAxOTII-*Rag2*[−/−] recipients. Three days after *i.v.* adoptive transfer (AT), the thymic entry of DCs and macrophages of CD45.1 donor origin was analysed. SL-TBI: sublethal total body irradiation. **f–h** Flow cytometry profiles, frequencies and numbers of CD45.1 total donor cells (**f**) as well as cDCs, pDCs (**g**) and macrophages (**h**) of CD45.1 donor origin in the thymus from OTII-*Rag2*[−/−] and RipmOVAxOTII-*Rag2*[−/−] recipients. Control: non-injected irradiated OTII-*Rag2*[−/−] mice. Data are representative of three independent experiments (*n* = 3–4 mice per group and per experiment). **d**, **h** MΦ: macrophage. Error bars show mean ± SEM, *\*p* < 0.05, *\*\*p* < 0.01, *\*\*\*p* < 0.001, *\*\*\*\*p* < 0.0001 using unpaired Student's *t*-test

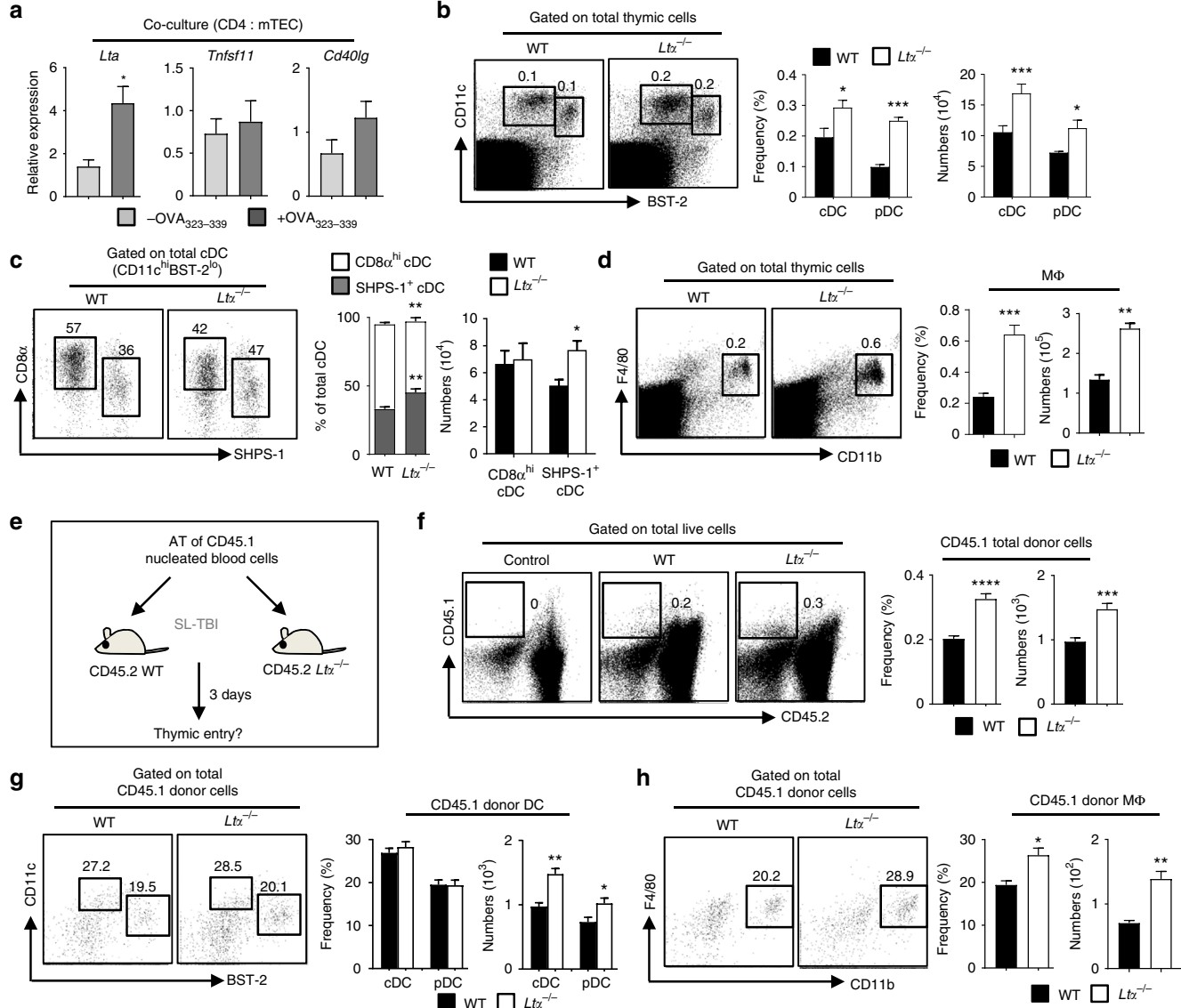

**Fig. 2** Enhanced migration of peripheral DCs and macrophages in the thymus of $Lt\alpha^{-/-}$ mice. **a** *Lta* (LTa), *Tnfsf11* (RANKL) and *Cd40lg* (CD40L) mRNAs were measured by qPCR in OTII CD4$^+$ thymocytes co-cultured with purified WT mTECs (CD45$^-$Ep-CAM$^+$BP-1$^{lo}$UEA-1$^+$) loaded ($n = 7$) or not ($n = 7$) with OVA$_{323-339}$ peptide. Data are representative of two independent experiments. **b**–**d** Flow cytometry profiles, frequencies and numbers of cDCs (CD11c$^{hi}$BST-2$^{lo}$), pDCs (CD11c$^{int}$BST-2$^{hi}$) (**b**), resident cDCs (CD8α$^{hi}$SHPS-1$^-$), migratory cDCs (CD8α$^{lo}$SHPS-1$^+$) (**c**) and macrophages (F4/80$^+$CD11b$^+$) (**d**) in the thymus from WT and $Lt\alpha^{-/-}$ mice. Data are representative of five independent experiments ($n = 3-4$ mice per group and per experiment). **e** Experimental setup: nucleated blood cells from CD45.1 WT congenic mice were adoptively transferred into sublethally irradiated CD45.2 WT and $Lt\alpha^{-/-}$ recipients. Thymic entry of DCs and macrophages of CD45.1 origin was analysed three days later. SL-TBI: sublethal total body irradiation. **f**–**h** Flow cytometry profiles, frequencies and numbers of CD45.1 total donor cells (**f**) as well as cDCs, pDCs (**g**) and macrophages (**h**) from CD45.1 donor origin in the thymus of WT and $Lt\alpha^{-/-}$ recipients. **f** Control: non-injected irradiated WT mice. **f**–**h** Data are representative of two independent experiments ($n = 3-4$ mice per group and per experiment). **d**, **h** MΦ: macrophage. Error bars show mean ± SEM, *$p < 0.05$, **$p < 0.01$, ***$p < 0.001$, ****$p < 0.0001$ using unpaired Student's *t*-test

(LTα) was upregulated in OTII CD4$^+$ thymocytes co-cultured with OVA$_{323-339}$-loaded mTECs (Fig. 2a), suggesting that LTα could be involved in the thymic entry of peripheral APCs mediated by mTEC-CD4$^+$ thymocyte crosstalk. To investigate this hypothesis, we first evaluated on thymic sections the area occupied by CD11c$^+$ cells in the cortex and medulla and found that DC enrichment in the medulla was increased in $Lt\alpha^{-/-}$ mice (Supplementary Fig. 6a). Consistently, flow cytometry analyses revealed that $Lt\alpha^{-/-}$ mice had increased frequencies and numbers of thymic cDCs and pDCs compared with wild-type (WT) mice (Fig. 2b). Furthermore, although numbers

of CD8α$^{hi}$SHPS-1$^-$ resident cDCs were unaltered, numbers of CD8α$^{lo}$SHPS-1$^+$ migratory cDCs were increased in these mice (Fig. 2c). Frequencies and numbers of F4/80$^+$CD11b$^+$ macrophages were also higher in $Lt\alpha^{-/-}$ thymi than in WT thymi (Fig. 2d), with a distribution largely increased in the medulla (Supplementary Fig. 6b). In line with a previous study[32], we observed that CD8α$^{lo}$SHPS-1$^+$ cDCs and pDCs were more proliferative than CD8α$^{hi}$SHPS-1$^-$ cDCs (Supplementary Fig. 6c). Similar frequencies of proliferating Ki-67$^+$ SHPS-1$^+$ cDCs, pDCs and macrophages in WT and $Lt\alpha^{-/-}$ mice indicate that the elevated numbers of these cell types in $Lt\alpha^{-/-}$ mice were not due to

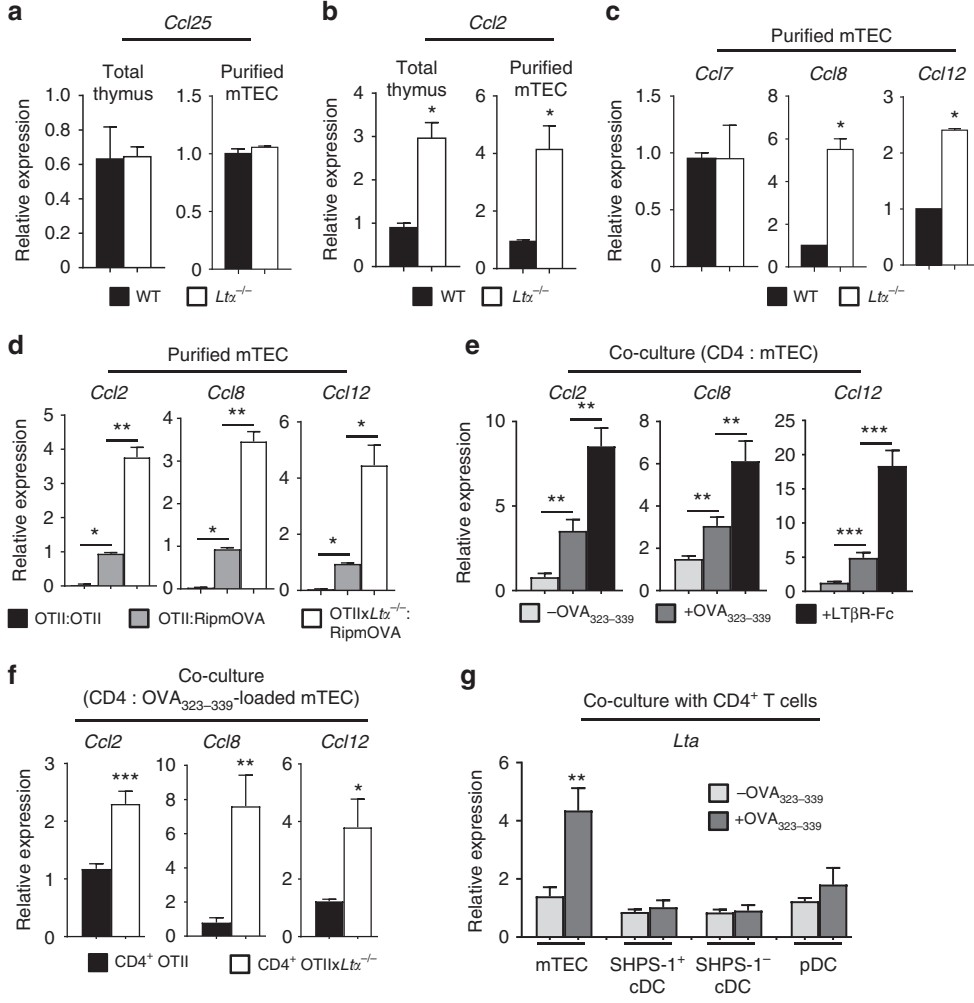

**Fig. 3** LTα negatively regulates CCL2, CCL8 and CCL12 expression in mTECs during crosstalk with CD4[+] thymocytes. **a–b** *Ccl25* (**a**) and *Ccl2* (**b**) mRNAs were measured by qPCR in the total thymus and in purified mTECs (CD45[-]Ep-CAM[+]BP-1[lo]UEA-1[+]) from WT ($n = 5$) and *Ltα*[−/−] ($n = 5$) mice. **c** *Ccl7*, *Ccl8* and *Ccl12* mRNAs were measured by qPCR in purified mTECs from WT ($n = 5$) and *Ltα*[−/−] ($n = 5$) mice. **d** *Ccl2*, *Ccl8* and *Ccl12* mRNAs were measured by qPCR in purified mTECs from OTII:OTII ($n = 6$), OTII:RipmOVA ($n = 6$) and OTIIx*Ltα*[−/−]:RipmOVA ($n = 6$) chimeras. **e** *Ccl2*, *Ccl8* and *Ccl12* mRNAs were measured by qPCR in purified mTECs loaded ($n = 6$) or not ($n = 6$) with OVA[323–339] peptide co-cultured with OTII CD4[+] thymocytes in the presence or not of recombinant LTβR-Fc chimera. **f** *Ccl2*, *Ccl8* and *Ccl12* mRNAs were measured by qPCR in purified mTECs loaded with OVA[323–339] peptide and co-cultured with CD4[+] thymocytes from OTII-*Rag2*[−/−] ($n = 9$) or OTII-*Rag2*[−/−]x*Ltα*[−/−] ($n = 9$) mice. **g** *Lta* mRNA was measured by qPCR in purified OTII CD4[+] thymocytes co-cultured with mTECs ($n = 7$), thymic SHPS-1[+] cDCs ($n = 8$), SHPS-1[-] cDCs ($n = 8$) and pDCs ($n = 8$) loaded or not with OVA[323–339] peptide. **a–g** Data are representative of two independent experiments. Error bars show mean ± SEM, *$p < 0.05$, **$p < 0.01$, ***$p < 0.001$ using two-tailed Mann–Whitney test for **b–d**, and unpaired Student's $t$-test for **e–g**

an increased cell proliferation, suggesting a role for LTα in repressing thymus homing of peripheral DCs and macrophages. To demonstrate this hypothesis, sublethally irradiated WT and *Ltα*[−/−] recipients were adoptively transferred with CD45.1 nucleated blood cells (Fig. 2e). Three days later, we found increased frequencies and numbers of CD45.1 donor cells in *Ltα*[−/−] thymi (Fig. 2f) that contained increased numbers of cDCs, pDCs and macrophages (Fig. 2g, h). Similar results were obtained by adoptively transferring CD45.1 splenic DC and macrophage-enriched cells into nonirradiated WT and *Ltα*[−/−] recipients (Supplementary Fig. 7). Accordingly, a substantial reduction in numbers of circulating SHPS-1[+] cDC, pDC and macrophage was observed in the blood of *Ltα*[−/−] mice ($p < 0.001$ for SHPS-1[+] cDCs, $p < 0.01$ for pDCs and $p < 0.05$ for macrophages by unpaired Student's $t$-test, Supplementary Fig. 8a–c). Consistently with previous observations[33], these cell types were also reduced in the spleen of these mice (Supplementary Fig. 8d–f). This demonstrates that increased numbers of DCs and macrophages

are limited to the thymus and result from an enhanced recruitment, indicating that LTα constitutes a feedback repressor of APC thymic entry.

We next hypothesised that LTα could regulate the expression of key chemokines involved in the thymus homing of DCs and macrophages. Because CCR9 has been implicated in the thymic entry of peripheral pDCs[18], we first assessed the expression of its ligand *Ccl25*, which was unaltered in the total thymus and purified mTECs from *Ltα*[−/−] mice (Fig. 3a). Since we and others previously showed that CCL2 overexpression in the thymus leads to increased thymic pDC cellularity and that its receptor CCR2 is involved in thymic SHPS-1[+] cDC and pDC homeostasis[19,34,35], we hypothesised that LTα could modulate CCL2 expression. Strikingly, *Ccl2* expression was substantially higher in both total thymus ($p < 0.05$ by a two-tailed Mann–Whitney test) and purified mTECs ($p < 0.05$ by a two-tailed Mann-Whitney test) in *Ltα*[−/−] mice than in WT mice (Fig. 3b). CCL2 is a major ligand for CCR2, which has other potential ligands, such as CCL7,

CCL8 and CCL12[36,37]. Although CCL8 has been detected in the thymus[19], CCL7 and CCL12 have not been yet described to be expressed in this tissue. In contrast to *Ccl7*, we found that *Ccl8* and *Ccl12* expression was also increased in *Ltα*[−/−] mTECs (Fig. 3c). Considering that LTα is induced upon crosstalk (Fig. 2a), we next examined whether *Ccl2*, *Ccl8* and *Ccl12* expression in mTECs could be regulated by crosstalk with OTII CD4[+] thymocytes. The expression of these three ligands was increased in mTECs from OTII:RipmOVA mice compared with OTII:OTII mice (Fig. 3d), which was even more pronounced in OTII:RipmOVA mice backcrossed on a *Ltα*[−/−] background, indicating that LTα represses mTEC ability to express these chemokines upon crosstalk. To further determine the role of direct Ag-specific interactions with CD4[+] thymocytes in this chemokine expression, WT mTECs loaded or not with OVA$_{323-339}$ peptide were co-cultured with OTII CD4[+] thymocytes. *Ccl2*, *Ccl8* and *Ccl12* were upregulated in OVA$_{323-339}$-loaded mTECs compared with unloaded mTECs (Fig. 3e). Moreover, the addition of a soluble LTβR-Fc chimera, which blocks LTα1β2/LTβR interactions, resulted in a more pronounced upregulation of these chemokines, indicating that LTα1β2/LTβR axis acts as a negative regulator of these chemokines upon mTEC-CD4[+] thymocyte crosstalk. We also found higher levels of *Ccl2*, *Ccl8* and *Ccl12* in mTECs co-cultured with CD4[+] thymocytes from OTIIx*Ltα*[−/−] compared with those from OTII mice, suggesting that LTα, specifically in CD4[+] thymocytes, controls the expression of these chemokines (Fig. 3f). Moreover, although DCs can cross-present Ags expressed by mTECs[13,14], the chemokine upregulation observed in co-cultures with OTII CD4[+] thymocytes, in absence of DCs (Fig. 3e), indicates that interactions with CD4[+] thymocytes are sufficient to induce CCL2, CCL8 and CCL12 expression in mTECs. Using an antibody allowing the detection of MCP1–4 (i.e., CCL2, CCL7, CCL8 and CCL13, the latter being not expressed in mice), we found that in contrast to mTECs, thymic DC subsets did not express detectable levels of these chemokines in OTII-*Rag2*[−/−] and RipmOVAxOTII-*Rag2*[−/−] mice as well as in WT and *Ltα*[−/−] mice (Supplementary Fig. 9). This suggests that DCs do not possess the ability to attract peripheral APCs through the production of these chemokines. Finally, in contrast to OVA$_{323-339}$-loaded SHPS-1[+] cDCs, SHPS-1[−] cDCs and pDCs, only OVA$_{323-339}$-loaded mTECs were able to induce *Lta* expression in CD4[+] thymocytes, excluding a potential implication of DCs in the regulation of these chemokines through LTα induction (Fig. 3g). Altogether, these data show that LTα represses CCL2, CCL8 and CCL12 expression induced in mTECs upon crosstalk with CD4[+] thymocytes.

**LTα-regulated thymic entry of APCs depends on CCR2.** CCL2, CCL8 and CCL12 chemokines are known ligands for CCR2[36], and CCL8 is also a ligand for CCR1 and CCR5[36,37,38,39]. To investigate a potential involvement of these chemokine receptors in the thymic entry of peripheral DCs and macrophages, we first examined their expression in blood-derived SHPS-1[+] cDCs, pDCs and macrophages in WT and *Ltα*[−/−] mice. All these cell types significantly expressed CCR2 in these mice ($p < 0.05$ by two-tailed Mann–Whitney test, Supplementary Fig. 10), indicating that they possess the ability to migrate in a CCR2-dependent manner into the thymus. CCR1 and CCR5 were also weakly detectable in these three cell types, although macrophages had high levels of CCR1 in both mice. To determine CCR2, CCR1 and CCR5 contributions in regulating the thymic pool of SHPS-1[+] cDCs, pDCs and macrophages, we generated mixed BM chimeras, in which lethally irradiated CD45.1xCD45.2 WT recipients were reconstituted with CD45.1 WT BM cells together

with either CD45.2 WT, *Ccr2*[RFP/RFP], *Ccr1*[−/−] or *Ccr5*[−/−] BM cells (ratio 1:1) (Fig. 4a). Six weeks later, thymic SHPS-1[+] cDCs, pDCs and macrophages were analysed (Supplementary Fig. 11). Strikingly, we found strongly reduced frequencies of these three cell types derived from CD45.2 *Ccr2*[RFP/RFP] BM cells compared with those derived from CD45.2 WT BM cells (Fig. 4b). While macrophages were unaffected, we also found a slight reduction in SHPS-1[+] cDCs and pDCs derived from CD45.2 *Ccr1*[−/−] BM and in these three cell types derived from CD45.2 *Ccr5*[−/−] BM cells. These results indicate that compared with CCR1 and CCR5, CCR2 is a key regulator of the thymic pool of SHPS-1[+] cDCs, pDCs and macrophages.

Finally, to firmly demonstrate that enhanced thymic entry of DCs and macrophages in *Ltα*[−/−] mice was mediated by CCR2, blood nucleated cells from CD45.1 WT mice and *Ccr2*[RFP/RFP]-deficient mice (ratio 1:1) were co-transferred into sublethally irradiated WT and *Ltα*[−/−] recipients (Fig. 4c). Consistently with our adoptive transfer (AT) experiments in *Ltα*[−/−] mice (Fig. 2e, f), frequencies and numbers of CD45.2[−]RFP[−] cells corresponding to CD45.1 donor cells (Supplementary Fig. 12a) were more elevated in *Ltα*[−/−] than in WT recipients (Fig. 4d). Among total donor cells, we found increased frequencies and numbers of cDCs, pDCs and macrophages (Supplementary Fig. 12b, c). In contrast, increased thymus homing of total donor cells and of these three cell types in *Ltα*[−/−] mice was strongly impaired when donor cells were of *Ccr2*[RFP/RFP]-deficient origin (Fig. 4d, e). Altogether, these data reveal that LTα controls the thymus homing of APCs in a CCR2-dependent manner.

**Induction of CCR2 ligands and NF-κB subunits in *Ltα*[−/−] mice.** We next investigated by which mechanisms CCL2, CCL8 and CCL12 chemokines are overexpressed in *Ltα*[−/−] mTECs. This cell type can be subdivided into two main subsets based on CD80 level[40] (Fig. 5a). Consistently with previous studies[41,42], *Ltα*[−/−] mice have normal frequencies and numbers of CD80[lo] (mTEC[lo]) and CD80[hi] (mTEC[hi]) mTECs, suggesting that increased expression in these chemokines (Fig. 3b, c) was not due to increased mTEC numbers. We found, by qPCR and flow cytometry, that CCR2 ligands were specifically upregulated in *Ltα*[−/−] mTEC[lo] (Fig. 5b, c). CCL2 and CCL8 are known to be regulated by the classical NF-κB pathway in different cell types[24–26]. Notably, p65 binding to the mouse *Ccl2* promoter is involved in CCL2 expression[43,44]. We identified two putative NF-κB binding sites for c-Rel and p65, by in silico analysis, in the *Ccl12* promoter region (Supplementary Table 1), suggesting that this gene could be also regulated by the classical NF-κB pathway. The level of p65 phosphorylation at serine 536 (ser536), which is associated with the upregulation of CCL2[45,46], was unaltered in *Ltα*[−/−] mTEC[lo] (Fig. 5d). We next assessed whether *Ltα*[−/−] mTEC[lo] have a differential usage in the classical and non-classical NF-κB pathways, the latter known to be preferentially induced by LTα1β2/LTβR axis[6,47]. We found at mRNA and protein levels that the non-classical NF-κB subunit *Relb* (RelB) was decreased whereas classical NF-κB subunits *Rel* (cRel) and *Rela* (p65) were enhanced in *Ltα*[−/−] mTEC[lo] (Fig. 5e–g). We next analysed the effect of LTα1β2/LTβR axis upon Ag-specific interactions with CD4[+] thymocytes in the regulation of NF-κB subunits in mTEC[lo] that express the LTβR receptor (Fig. 5h). Interestingly, mTECs co-cultured with OTIIx*Ltα*[−/−] CD4[+] thymocytes had reduced levels of *Relb* compared with mTECs co-cultured with OTII CD4[+] thymocytes (Fig. 5i). In contrast, increased expression of *Rel* and *Rela* correlates with CCL2, CCL8 and CCL12 overexpression in these cells (Fig. 3f, Fig. 5i). Thus, the disruption of the LTα1β2/LTβR axis in the

context of Ag-specific interactions with CD4$^+$ thymocytes leads to the upregulation of cRel and p65 classical NF-κB subunits and CCL2, CCL8 and CCL12 chemokines, suggesting that the chemokine upregulation in $Lt\alpha^{-/-}$ mTECs is controlled by the overexpression of classical NF-κB subunits.

**LTα-regulated APC thymic entry fine-tunes clonal deletion.** Since thymus homing of peripheral DCs and macrophages was enhanced in $Lt\alpha^{-/-}$ mice (Fig. 2e–h), we next investigated its

impact on clonal deletion. Interestingly, numbers of DP (CD4$^+$CD8$^+$), CD4$^{lo}$CD8$^{lo}$ and CD4$^+$ SP (CD4$^+$CD8$^-$) cells were significantly reduced in $Lt\alpha^{-/-}$ mice compared to WT mice (Fig. 6a, b). We also observed reduced numbers of CCR7$^-$ cortical and CCR7$^+$ medullary CD4$^+$ SP (Fig. 6c). This cannot be explained by defective thymus homing of early thymic progenitors, since we previously observed that their numbers were normal in $Lt\alpha^{-/-}$ thymi[42]. We thus analysed strongly autoreactive thymocytes, based on the expression of the Ikaros family transcription factor Helios and PD-1, as previously described[8]

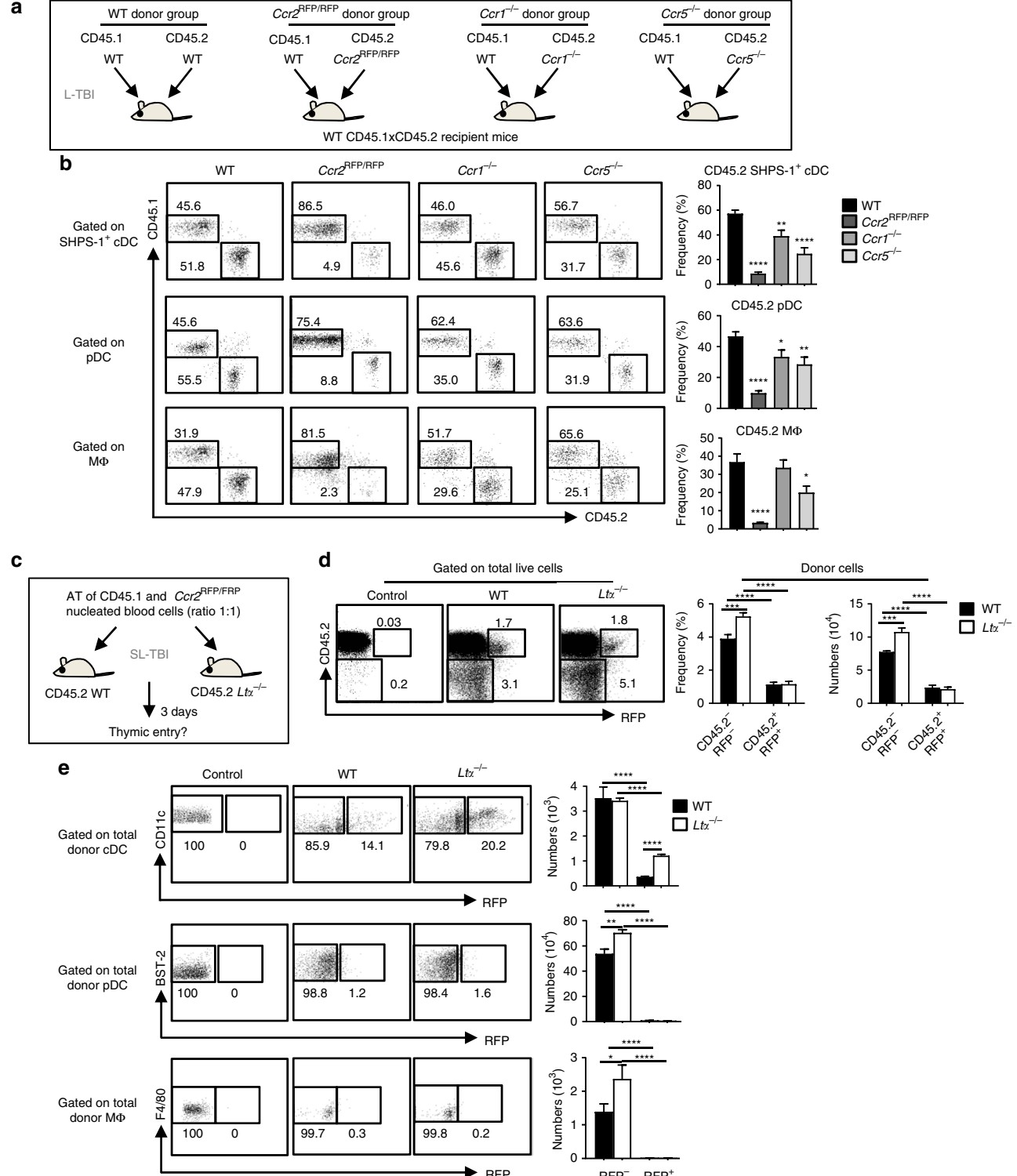

(Fig. 6a). We found reduced numbers of autoreactive Helios[+] PD-1[−] cells in CCR7[−] and CCR7[+] CD4[+] SP cells in $Lt\alpha^{-/-}$ mice, suggesting that clonal deletion was enhanced both in the cortex and medulla in these mice (Fig. 6d, e). Furthermore, we found reduced numbers of cortical Helios[+]PD-1[+]CD4[lo]CD8[lo] post-positively selected cells (Fig. 6f). Similar results were observed when $Lt\alpha^{-/-}$ BM cells were transplanted into CD45.1 WT recipients (Supplementary Fig. 13), suggesting that LTα expression in haematopoietic cells controls the deletion of cortical and medullary thymocytes. An enhanced clonal deletion in $Lt\alpha^{-/-}$ mice is supported by reduced frequencies and numbers of CD69[−]CD62L[+] mature CD4[+] SP in these mice (Fig. 6g). Therefore, these data suggest that enhanced thymus homing of DCs and macrophages in $Lt\alpha^{-/-}$ mice leads to increased clonal deletion.

To confirm that LTα-regulated thymic entry of DCs and macrophages impacts the negative selection, OTII-$Rag2^{-/-}$ mice were backcrossed on a $Lt\alpha^{-/-}$ background. Similarly to $Lt\alpha^{-/-}$ mice (Fig. 3b, c), Ccl2, Ccl8 and Ccl12 were upregulated in mTECs from OTII-$Rag2^{-/-}$x$Lt\alpha^{-/-}$ compared to OTII-$Rag2^{-/-}$ mice (Fig. 7a). Consistent with our findings that LTα regulates DC and macrophage thymic cellularity (Fig. 2b–d), numbers of SHPS-1[+] cDCs, pDCs and macrophages were higher in the thymus of these mice (Fig. 7b). Furthermore, thymi of OTII-$Rag2^{-/-}$x$Lt\alpha^{-/-}$ recipients previously transferred with CD45.1 DC and macrophage-enriched cells contained increased numbers of donor SHPS-1[+] cDCs, pDCs and macrophages (Fig. 7c–e), confirming that the thymic migration of APCs is favoured on a $Lt\alpha^{-/-}$ background. Furthermore, we found that the AT of OVA$_{323–339}$-loaded DC and macrophage-enriched cells was effective at eliminating OTII DP and Vα2[+]Vβ5[+]CD4[+] SP cells compared with unloaded APCs (Fig. 7f, g). This phenomenon was increased in OTII-$Rag2^{-/-}$x$Lt\alpha^{-/-}$ mice, confirming an enhanced negative selection on a $Lt\alpha^{-/-}$ background. These data thus firmly demonstrate that LTα-regulated APC thymic entry controls the clonal deletion of autoreactive thymocytes.

**High ability of cDCs and macrophages for clonal deletion**. We next assessed the tissue distribution of SHPS-1[+] cDCs, pDCs and macrophages by transferring cells sorted from CCR2[RFP/+] heterozygous mice (Fig. 8a), allowing us to track their thymic entry with the red fluorescent protein (RFP) reporter gene. For better detection, cell-sorted peripheral DCs and macrophages from CCR2[RFP/+] donor mice were adoptively transferred into $Lt\alpha^{-/-}$ recipients, in which thymus homing of these cells is increased (Fig. 2e–h, Supplementary Fig. 14a, b). Consistent with our AT experiments of CD45.1 donor cells (Figs. 1e–h, 2e–h, 4c–e, 7d, e), we found that these three cell types efficiently homed into the $Lt\alpha^{-/-}$ thymus and retained their phenotypic hallmarks, as revealed by CD11c, BST-2 and F4/80 staining (Fig. 8b). Interestingly, RFP[+] cDCs and pDCs were

preferentially located in the cortex, whereas RFP[+]F4/80[+] macrophages were similarly distributed in the cortex and medulla.

To investigate the respective role of these three cell types in clonal deletion, WT BM-derived cDCs, pDCs and macrophages, expressing CCR2 and MHCII molecules, were generated (Fig. 8c, Supplementary Fig. 14c–e). This indicates that these cell types should be competent to migrate in a CCR2-dependent manner and present Ags via MHCII molecules. The same number of OVA$_{323–339}$-loaded BM-derived cDCs, pDCs or macrophages was first adoptively transferred into OTII-$Rag2^{-/-}$ mice (Fig. 8d). OVA$_{323–339}$-loaded BM-derived cDCs and pDCs were able to delete autoreactive thymocytes in OTII-$Rag2^{-/-}$ mice compared to non-injected OTII-$Rag2^{-/-}$ controls (Fig. 8e). Whereas the role of thymic macrophages in clonal deletion remains largely elusive, we found that OVA$_{323–339}$-loaded BM-derived macrophages were able to delete in vivo both DP and Vα2[+]Vβ5[+]CD4[+] SP thymocytes (Fig. 8e). This is consistent with the observation that adoptively transferred macrophages were localised in both the cortex and the medulla (Fig. 8b), where DP and SP thymocytes are respectively eliminated[7]. Importantly, BM-derived cDCs and macrophages were more efficient than BM-derived pDCs in deleting total thymocytes, including DP and Vα2[+]Vβ5[+]CD4[+] SP cells. Consistently, BM-derived cDCs and macrophages expressed higher levels of MHCII than BM-derived pDCs (Fig. 8c). Furthermore, thymic SHPS-1[+] cDCs express higher levels of MHCII than pDCs, and are thus more prone to clonal deletion[20]. OVA$_{323–339}$-loaded BM-derived cDCs, pDCs and macrophages deleted more efficiently total thymocytes, DP and Vα2[+]Vβ5[+]CD4[+] SP cells in OTII-$Rag2^{-/-}$x $Lt\alpha^{-/-}$ than in OTII-$Rag2^{-/-}$ mice (Fig. 8e), which is consistent with the superior ability of these three cell types to home into the thymus on a $Lt\alpha^{-/-}$ background (Fig. 2e–h, Fig. 7d, e). Altogether these data indicate that migratory cDCs and macrophages have a higher capacity to delete autoreactive thymocytes than pDCs, a phenomenon accentuated in $Lt\alpha^{-/-}$ mice.

## Discussion

Thymic clonal deletion crucially prevents the generation of hazardous autoreactive T cells that could induce autoimmunity. mTECs are essential in this process through their ability to widely express self-Ags that can be cross-presented by resident cDCs[13–15,48,49]. Peripheral DCs, by continuously migrating into the thymus, also contribute to the deletion of autoreactive thymocytes by sampling peripheral self-Ags[17–19]. Although migratory DCs are involved in this tolerogenic process, the implication of mTEC-thymocyte crosstalk in regulating their thymic recruitment remained unknown so far.

We provide strong evidence that Ag-specific interactions between mTECs and CD4[+] thymocytes regulate the thymic entry of peripheral DCs and macrophages. We found increased numbers of SHPS-1[+] cDCs, pDCs and macrophages in the

**Fig. 4** Crucial role of CCR2 in the recruitment of circulating cDCs, pDCs and macrophages into the thymus. **a** Experimental setup: lethally irradiated CD45.1xCD45.2 WT recipients were reconstituted with BM cells from CD45.1 WT mice together with CD45.2 WT (WT donor group), CD45.2 Ccr2[RFP/RFP] deficient (Ccr2[RFP/RFP] donor group), CD45.2 Ccr1[−/−] (Ccr1[−/−] donor group) or CD45.2 Ccr5[−/−] (Ccr5[−/−] donor group) mice (ratio 1:1). Thymic DCs and macrophages were analysed six weeks later. L-TBI: lethal total body irradiation. **b** Flow cytometry profiles and frequencies of thymic SHPS-1[+] cDCs, pDCs and macrophages of CD45.2 origin in each group of mice. **a**, **b** Data are representative of two independent experiments (n = 4–5 mice per group and per experiment). **c** Experimental setup: nucleated blood cells from CD45.1 WT congenic mice and Ccr2[RFP/RFP] deficient mice were adoptively transferred into sublethally irradiated CD45.2 WT or $Lt\alpha^{-/-}$ recipients (ratio 1:1). Thymic entry of donor DCs and macrophages was analysed three days later. SL-TBI: sublethal total body irradiation. **d** Flow cytometry profiles, frequencies and numbers of CD45.2[+]RFP[+] and CD45.2[−]RFP[−] total donor cells in the thymus of WT and $Lt\alpha^{-/-}$ recipients. **e** Flow cytometry profiles and numbers of SHPS-1[+] cDCs, pDCs and macrophages of CD45.2[−]RFP[−] or CD45.2[+]RFP[+] donor origin. **c–e** Data are representative of three independent experiments (n = 4 mice per group and per experiment). **b**, **e** MΦ: macrophage. Error bars show mean ± SEM, *p < 0.05, **p < 0.01, ***p < 0.001, ****p < 0.0001 using unpaired Student's t-test

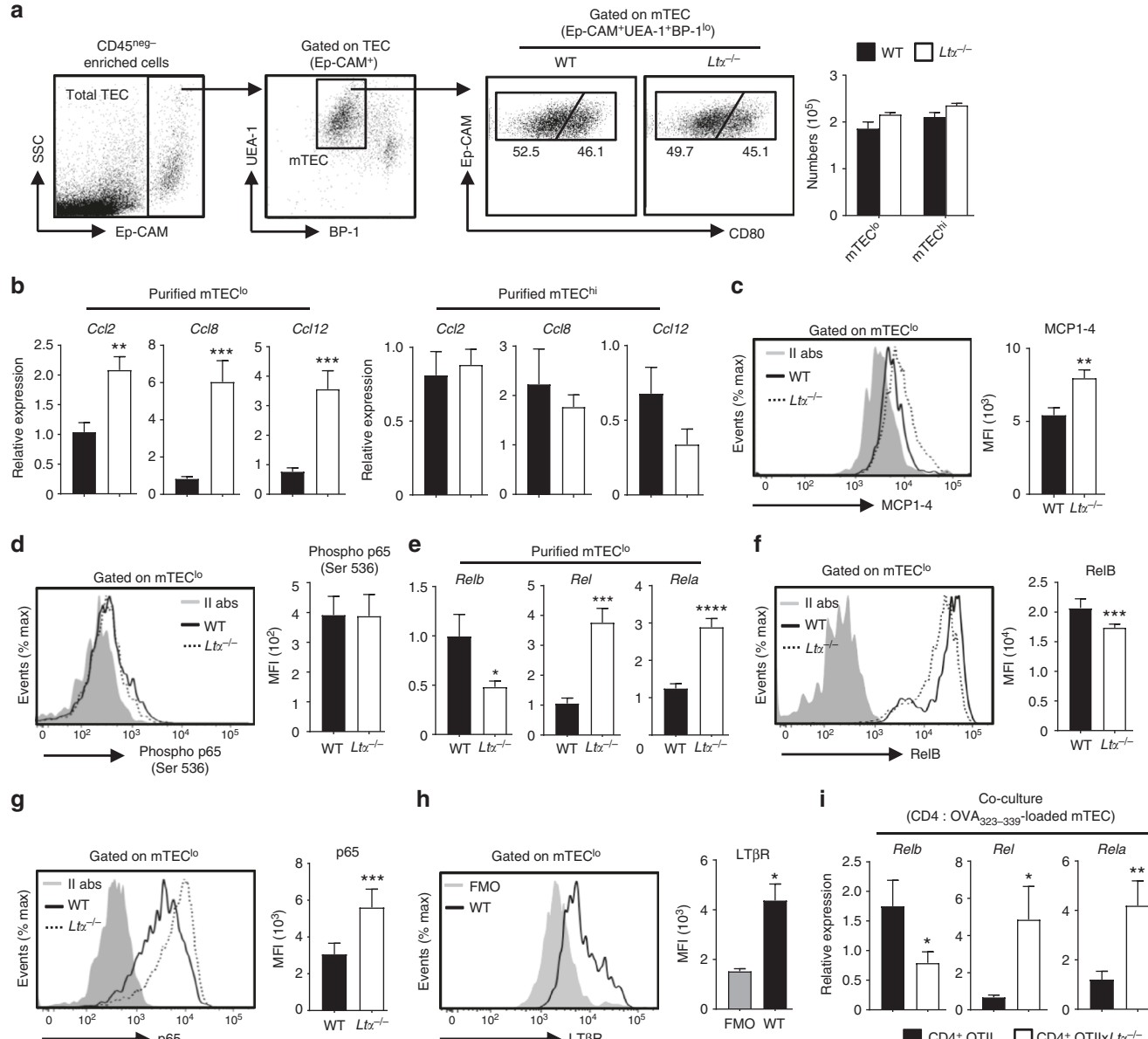

**Fig. 5** Upregulation of CCL2, CCL8 and CCL12 chemokines, specifically in $Lt\alpha^{-/-}$ mTEC$^{lo}$, correlates with the upregulation of classical NF-κB subunits. **a** Gating strategy and flow cytometry profiles used to analyse total mTECs (CD45$^-$Ep-CAM$^+$BP-1$^{lo}$UEA-1$^+$), mTEC$^{lo}$ (CD45$^-$Ep-CAM$^+$BP-1$^{lo}$UEA-1$^+$ CD80$^{lo}$) and mTEC$^{hi}$ (CD45$^-$Ep-CAM$^+$BP-1$^{lo}$UEA-1$^+$CD80$^{hi}$) in WT and $Lt\alpha^{-/-}$ mice. This gating strategy was used to sort and analyse mTECs in Figs. 3 and 5. The histogram shows numbers of mTEC$^{lo}$ and mTEC$^{hi}$ cells in both mice. **b** Ccl2, ccl8 and ccl12 mRNAs were measured by qPCR in purified mTEC$^{lo}$ and mTEC$^{hi}$ from WT ($n = 8$) and $Lt\alpha^{-/-}$ ($n = 8$) mice. **c**, **d** MCP1–4 (**c**) and phosphorylation of Ser536 p65 (**d**) were analysed by flow cytometry in mTEC$^{lo}$ from WT and $Lt\alpha^{-/-}$ mice. Histograms show the MFI of MCP1–4 and phospho p65 (Ser536). **e** Relb, Rel and Rela mRNAs were measured by qPCR in purified mTEC$^{lo}$ from WT ($n = 8$) and $Lt\alpha^{-/-}$ ($n = 8$) mice. **f**, **g** RelB (**f**) and p65 (**g**) protein levels were analysed by flow cytometry in mTEC$^{lo}$ from WT and $Lt\alpha^{-/-}$ mice. Histograms show the MFI of RelB and p65 expression. **h** LTβR expression was analysed by flow cytometry in WT mTEC$^{lo}$. The histogram shows the MFI of LTβR. **i** Relb, Rel and Rela mRNAs were measured by qPCR in mTECs loaded ($n = 6$) or not ($n = 6$) with OVA$_{323-339}$ peptide co-cultured with CD4$^+$ thymocytes from OTII-$Rag2^{-/-}$ mice or OTII-$Rag2^{-/-}$x$Lt\alpha^{-/-}$ mice. **a**–**i** Data are representative of two independent experiments ($n = 3$-4 mice per group and per experiment). II abs secondary antibodies, FMO fluorescence minus one, MFI mean fluorescence intensity. Error bars show mean ± SEM, *$p < 0.05$, **$p < 0.01$, ***$p < 0.001$, ****$p < 0.0001$ using one-tailed Mann–Whitney test for **c**, **f**, **g**, two-tailed Mann–Whitney test for **h** and unpaired Student's $t$-test for **b**, **e** and **i**.

thymus of RipmOVAxOTII-$Rag2^{-/-}$ and OTII:RipmOVA mice compared with OTII-$Rag2^{-/-}$ and OTII:OTII mice, respectively. Furthermore, thymus homing of these cell types was enhanced in RipmOVAxOTII-$Rag2^{-/-}$ recipients upon AT of donor cells. Although RANKL and CD40L are implicated in mTEC-thymocyte crosstalk[12], these two TNF members were expressed at similar extents in CD4$^+$ thymocytes from OTII-$Rag2^{-/-}$ and RipmOVAxOTII-$Rag2^{-/-}$ mice, indicating that they are unlikely

responsible for the increased thymic entry of peripheral APCs observed in RipmOVAxOTII-$Rag2^{-/-}$ mice. Nevertheless, we cannot exclude a potential role of these TNF members in other aspects of thymic DC biology. Future investigations are expected to clarify this issue. We show that this regulatory mechanism of peripheral APC recruitment is tightly controlled by LTα, which is specifically induced in autoreactive CD4$^+$ thymocytes upon crosstalk with mTECs[50]. Of note, LTα was shown to be expressed

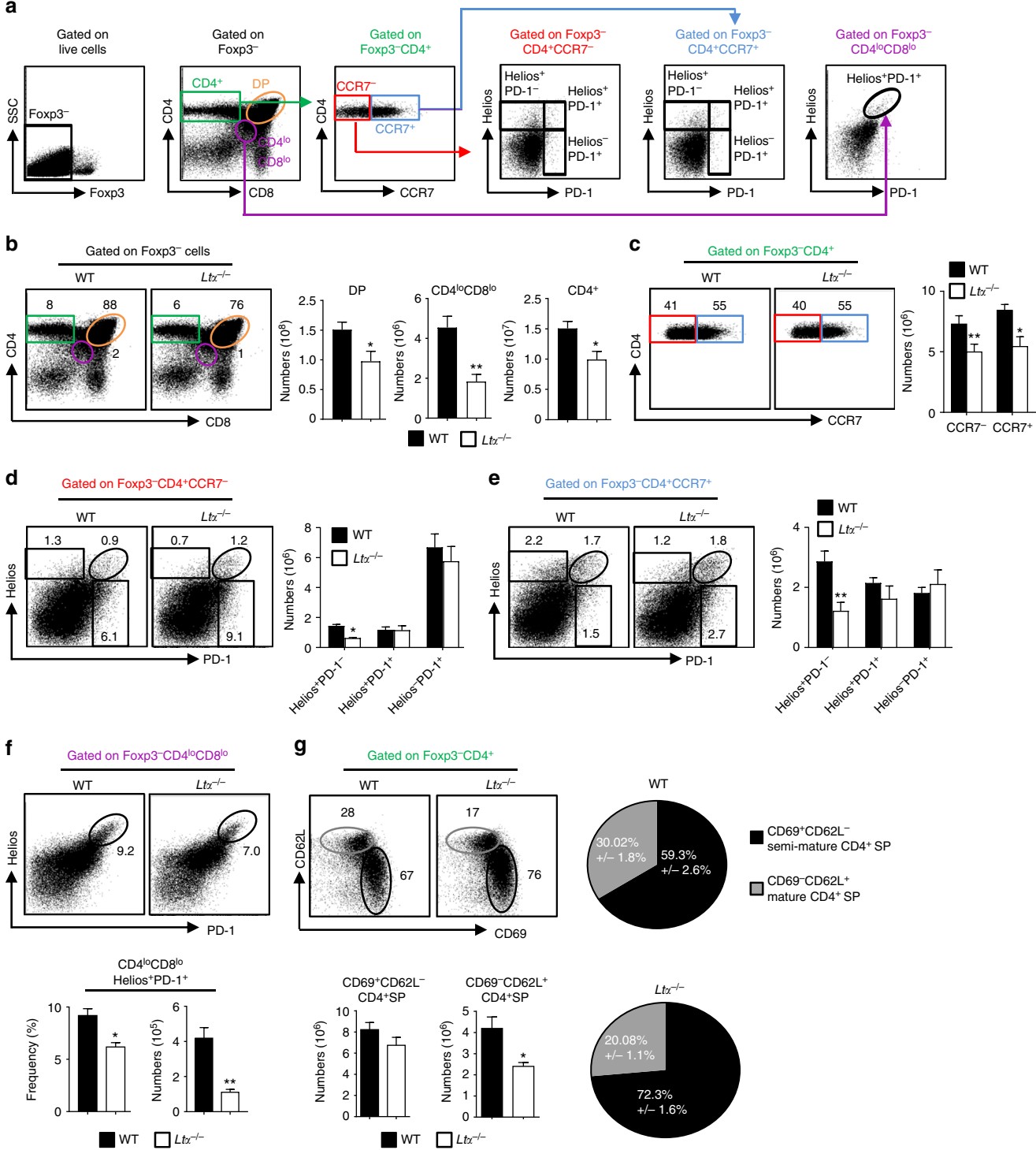

**Fig. 6** Enhanced clonal deletion in the thymus of $Lt\alpha^{-/-}$ mice. **a** Gating strategy used to analyse thymocyte subsets by flow cytometry. Foxp3$^-$ cells were analysed for CD4 and CD8 expression. Foxp3$^-$CD4$^+$ SP (CD4$^+$CD8$^-$) cells were analysed for the expression of CCR7. CCR7$^-$CD4$^+$, CCR7$^+$CD4$^+$ SP and CD4$^{lo}$CD8$^{lo}$ cells were analysed for the expression of Helios and PD-1. **b**, **c** Flow cytometry profiles and numbers of DP, CD4$^{lo}$CD8$^{lo}$ and CD4$^+$ SP thymocytes (**b**) and CCR7$^-$CD4$^+$ and CCR7$^+$CD4$^+$ SP cells (**c**). **d–f** Flow cytometry profiles and numbers of Helios$^+$PD-1$^-$, Helios$^+$PD-1$^+$ and Helios$^-$PD-1$^+$ cells in CD4$^+$CCR7$^-$ (**d**), CD4$^+$CCR7$^+$ (**e**) and CD4$^{lo}$CD8$^{lo}$ cells (**f**) in the thymus from WT and $Lt\alpha^{-/-}$ mice. **g** Flow cytometry profiles, frequencies and numbers of CD69$^+$CD62L$^-$ semi-mature and CD69$^-$CD62L$^+$ mature in Foxp3$^-$CD4$^+$ SP thymocytes from WT and $Lt\alpha^{-/-}$ mice. **b–g** Data are representative of two independent experiments ($n = 4$ mice per group and per experiment). Error bars show mean ± SEM, *$p < 0.05$, **$p < 0.01$ using one-tailed Mann–Whitney test

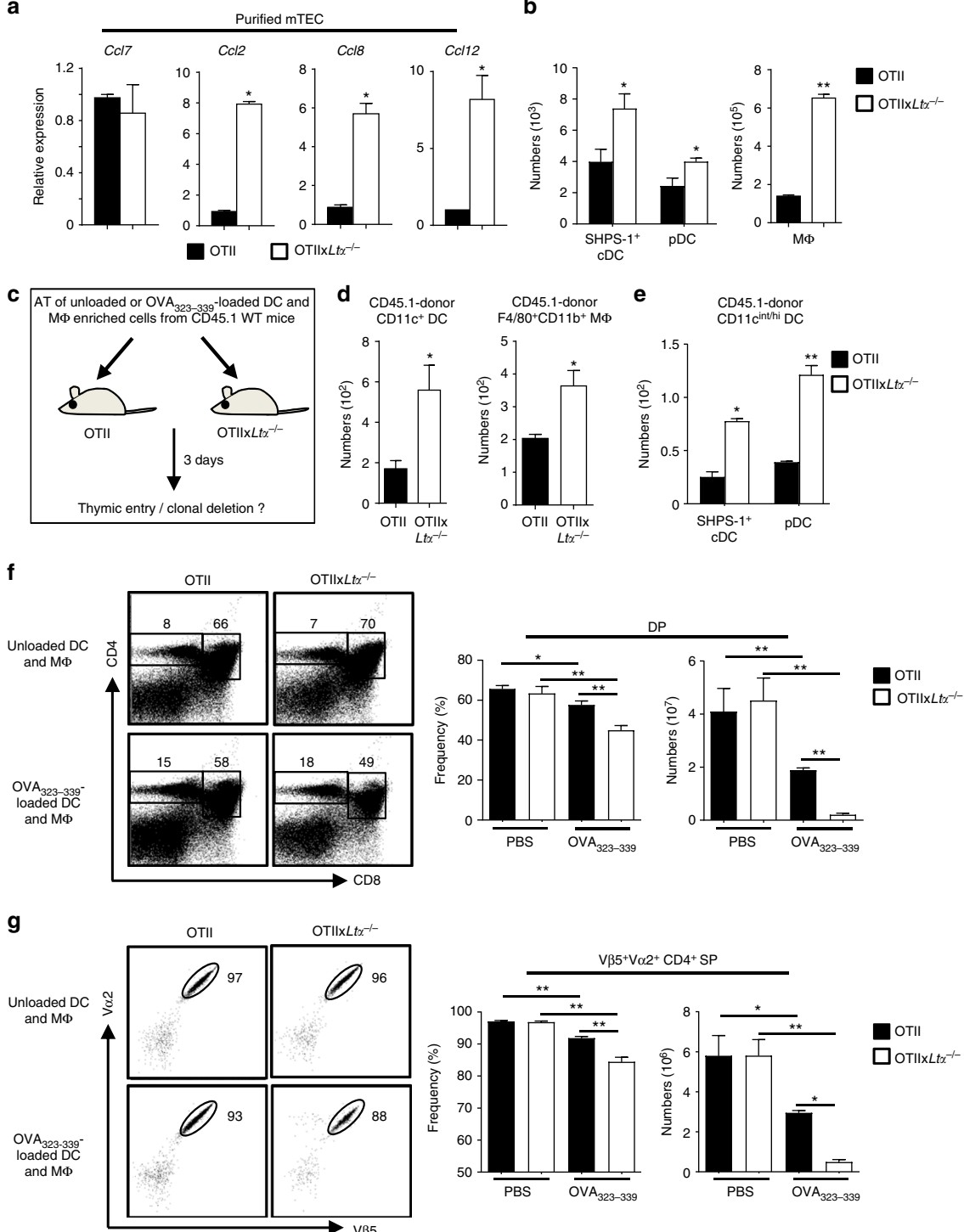

**Fig. 7** LTα-regulated thymic entry of DCs and macrophages leads to an enhanced clonal deletion. **a** *Ccl7*, *Ccl2*, *Ccl8* and *Ccl12* mRNAs were measured by qPCR in purified mTECs (CD45⁻Ep-CAM⁺BP-1ˡᵒUEA-1⁺) from OTII-*Rag2*⁻/⁻ (*n* = 3) and OTII-*Rag2*⁻/⁻x*Ltα*⁻/⁻ (*n* = 3) mice. Data are representative from two independent experiment (*n* = 1–2 mice per group). **b** Numbers of SHPS-1⁺ cDCs, pDCs and macrophages were analysed by flow cytometry in the thymus from OTII-*Rag2*⁻/⁻ and OTII-*Rag2*⁻/⁻x*Ltα*⁻/⁻ mice. **c** Experimental setup: AT of unloaded or OVA₃₂₃₋₃₃₉-loaded DC and macrophage-enriched cells from CD45.1 congenic mice into OTII-*Rag2*⁻/⁻ or OTII-*Rag2*⁻/⁻x*Ltα*⁻/⁻ recipients. Thymic entry of DCs and macrophages of CD45.1 origin was analysed three days later. **d**, **e** Numbers of CD45.1 donor cDCs, macrophages (**d**), SHPS-1⁺ cDCs and pDCs (**e**) were analysed in the thymus by flow cytometry. **f**, **g** Flow cytometry profiles, frequencies and numbers of DP (**f**) and Vβ5⁺Vα2⁺CD4⁺ SP (**g**) cells analysed three days after AT. **b**–**g** Data are representative from three to four independent experiments (*n* = 3–4 mice per group and per experiment). (**b**, **c**, **d**, **f**, **g**) MΦ: macrophage. Error bars show mean ± SEM, *p < 0.05, **p < 0.01, using one-tailed Mann–Whitney test

in SP thymocytes as a membrane anchored LTα1β2 hetero-complex, which binds to LTβR[30,47]. Our data show that LTα negatively regulates the expression of CCL2, CCL8 and CCL12 chemokines in mTECs, which attenuates thymus homing of APCs, mainly in a CCR2-dependent manner. Consequently, migratory cDCs, pDCs and macrophages were increased in $Lt\alpha^{-/-}$ compared to WT thymi. Contrarily to $Lt\alpha^{-/-}$ mice, $Ltbr^{-/-}$ mice have reduced numbers of thymic SHPS-1⁻ resident cDCs and pDCs[51]. It remains nevertheless unclear whether LTβR

controls the thymic pool of macrophages and peripheral APC entry. These differences are not surprising, since these mice have distinct defects in medulla organisation, mTEC subsets and autoimmunity[12]. Here, we demonstrate by AT of donor cells both in sublethally irradiated and unmanipulated mice, that LTα negatively regulates the thymus homing not only of peripheral cDCs and pDCs but also of macrophages.

Our results, based on OTII:OTII and OTII:RipmOVA mTECs and mTEC-CD4⁺ thymocyte co-cultures, strongly indicate that

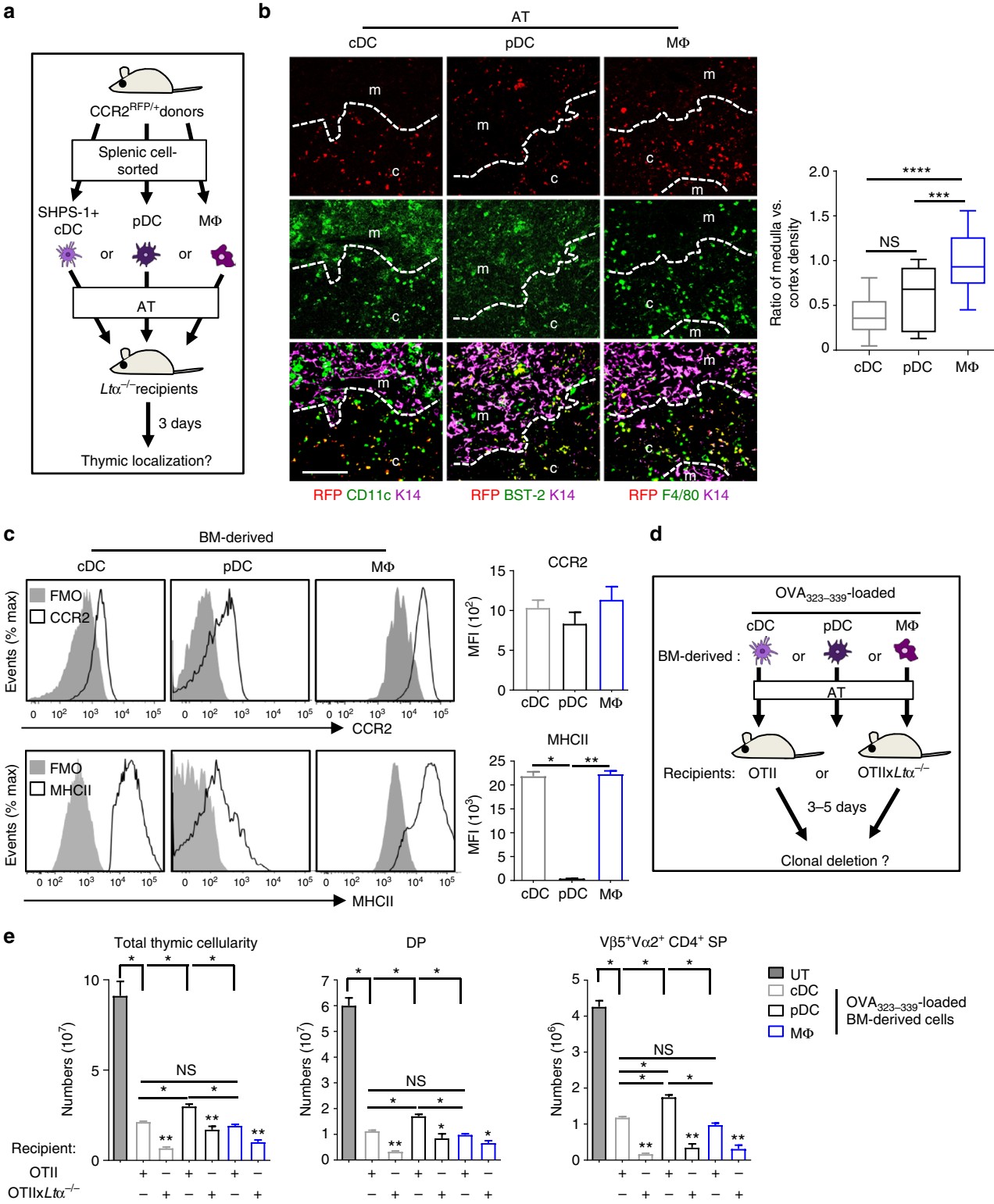

*Ccl2*, *Ccl8* and *Ccl12* were upregulated in mTECs upon Ag-specific interactions with CD4$^+$ thymocytes. This upregulation was negatively controlled by LTα, specifically in CD4$^+$ thymocytes, since it was exacerbated in absence of LTα or when LTα1β2/LTβR interactions were blocked. Furthermore, CCL2, CCL8 and CCL12 were specifically upregulated in *Ltα*$^{-/-}$ mTEC$^{lo}$. Although mTEC$^{lo}$ were initially described to contain precursors giving rise to functional mTEC$^{hi}$[12], they were reported to contain mTECs, expressing the chemokine CCL21 and the transcription factor FEZF2 implicated in the attraction of thymocytes into the medulla and in the expression of self-Ags, respectively[52,53]. Our study revealed that mTEC$^{lo}$ also express chemokines involved in the thymic entry of peripheral APCs, further highlighting a functional role of this subset in T cell selection. We further show that LTα controls mTEC$^{lo}$ properties by regulating CCL2, CCL8 and CCL12. This chemokine upregulation was likely due to a differential usage in the classical and non-classical NF-κB pathways, which are known to function as an interconnected signalling network rather than two independent pathways, even in mTECs[6,54,55]. Whereas in WT conditions, the lymphotoxin signalling activates preferentially the non-classical NF-κB pathway[6,47], our data indicate a preferential usage of the classical NF-κB pathway, characterised by an upregulation of cRel and p65 when the lymphotoxin signalling is disrupted. Since CCL2 and CCL8 are regulated by classical NF-κB members[24–26], which are overexpressed in *Ltα*$^{-/-}$ mice, this chemokine upregulation observed in *Ltα*$^{-/-}$ mTEC$^{lo}$ is likely mediated by the classical NF-κB pathway. Chromatin immunoprecipitation assays are expected to prove a direct regulation of these chemokines by classical NF-κB members. However, the mTEC$^{lo}$ subset represents a rare cell type (~$2 \times 10^5$ cells/thymus) rendering technically difficult such an approach. Furthermore, although DCs can cross-present Ags expressed by mTECs[13,14], it is unlikely that they participate in CCL2, CCL8 and CCL12 expression since co-cultures of CD4$^+$ thymocytes with mTECs alone was sufficient to induce these chemokines. Furthermore, DCs did not express CCR2 ligands in the context of mTEC-CD4$^+$ thymocyte crosstalk or upon LTα-mediated regulation. Another hypothesis would be that DCs could be indirectly involved in this regulation by inducing LTα in CD4$^+$ thymocytes. We showed that in contrast to mTECs, neither thymic SHPS-1$^+$, SHPS-1$^-$ cDCs nor pDCs were able to induce LTα in CD4$^+$ thymocytes, excluding a potential contribution of DCs in the expression of these chemokines through LTα upregulation. Nevertheless, we cannot definitively rule out a role of DC-mediated indirect antigen presentation in the thymic recruitment of peripheral APCs. Experiments based on the use of DC-depleted mice in the context of thymic crosstalk are expected to clarify this issue in the next future.

Mixed BM chimera and AT experiments demonstrated that CCR2 plays a major role in the thymus homing of SHPS-1$^+$ cDCs, pDCs and macrophages. Although a drastic reduction (~80–95%) of these three cell types was observed in the *Ccr2*$^{RFP/RFP}$-deficient donor group, a moderate but non-negligible decrease of thymic SHPS-1$^+$ cDCs, pDCs and macrophages was observed in *Ccr1*$^{-/-}$ and *Ccr5*$^{-/-}$ groups. While pDCs have been reported to migrate through CCR9[18], our data thus show that they also use CCR2. Moreover, co-AT of WT and *Ccr2*$^{RFP/RFP}$-deficient donor cells indicates that although the thymic entry of SHPS-1$^+$ cDCs is impaired in the absence of CCR2, this process is not fully abrogated, suggesting the implication of other chemokine receptor(s). Potential candidates could be CCR1 and CCR5, since thymic SHPS-1$^+$ cDCs were moderately disturbed in *Ccr1*$^{-/-}$ and *Ccr5*$^{-/-}$ donor groups. Future experiments, based on the analysis of *Ccr2* and *Ccr5* as well as *Ccr2* and *Ccr1* double-deficient mice, are expected to clarify this issue.

Interestingly, since negatively selected thymocytes do not directly die, but instead remain viable for few hours in the medulla[56], it is likely that autoreactive thymocytes have sufficient time to provide instructive signals to mTECs, that would regulate the thymic recruitment of peripheral DCs and macrophages. Interestingly, we demonstrate that this regulation loop controls the clonal deletion of autoreactive T cells (Supplementary Fig. 15). Autoreactive thymocytes were highly deleted at the DP, CD4$^{lo}$CD8$^{lo}$ and CD4$^+$ SP stages in *Ltα*$^{-/-}$ mice, indicating that enhanced clonal deletion occurs in both the cortex and medulla. By AT experiments, we demonstrated that cDCs, pDCs and macrophages were all able to delete efficiently DP and SP cells, a process accentuated on a *Ltα*$^{-/-}$ background. According to the type of negatively selected cells, SHPS-1$^+$ cDCs and pDCs were preferentially localised in the cortex, whereas macrophages were distributed throughout the thymus. Macrophages have been associated with the clearance of apoptotic bodies in the thymus[22,23]. Although they share common hallmarks with DCs, by expressing MHCII and CD80/CD86 molecules, implicated in T cell selection, their role in clonal deletion and the mechanisms that sustain their thymic entry remain unknown. So far, only one group has reported that F4/80$^+$CD11b$^+$ macrophages are able to delete autoreactive thymocytes in vitro by using reaggregated thymic organ cultures[57]. Here, we unravel that peripheral macrophages migrate in a CCR2-dependent manner into the thymus and that they may play an unsuspected role in clonal deletion.

Finally, from a therapeutic perspective, because *Ltα* deficiency increases DC and macrophage thymic entry, it would be interesting to determine whether LTα loss can protect and treat from autoimmunity. Generating an inducible transgenic mouse model allowing LTα deletion at a specific time point, i.e., before or after the development of autoimmune signs, would be useful to define the potential of LTα as a new target to prevent or treat autoimmunity associated with defective T cell selection.

In sum, this complex cellular interplay between mTECs, CD4$^+$ thymocytes, peripheral DCs and macrophages constitutes a fine-

**Fig. 8** Migratory cDCs and macrophages are more efficient than pDCs for clonal deletion, a process accentuated on a *Ltα*$^{-/-}$ background. **a** Experimental setup: *Ltα*$^{-/-}$ recipients were injected with cell-sorted cDCs, pDCs or macrophages from *Ccr2*$^{RFP/+}$ heterozygous mice. The thymic localisation of these cell types was analysed three days later on thymic sections. **b** Thymic sections were stained with antibodies against the medulla specific marker K14 (magenta) and CD11c (green), BST-2 (green) or F4/80 (green). Adoptively transferred RFP$^+$ cells were detected in red. m and c denote the medulla and the cortex, respectively. The graph shows the ratio of medullary vs. cortical density of adoptively transferred cells. Twenty-five sections derived from two mice for each genotype were quantified for each condition. Scale bar: 100 μm. **c** CCR2 and MHCII in BM-derived cDCs, pDCs and macrophages from WT mice were analysed by flow cytometry. Histograms show the MFI values of CCR2 and MHCII expression normalised to the FMO value of each population analysed. Data are representative of two independent experiments ($n = 3$ mice per group and per experiment). *FMO* fluorescence minus one, *MFI* mean fluorescence intensity. **d** Experimental setup: AT of purified OVA$_{323-339}$-loaded BM-derived cDCs, pDCs or macrophages into OTII-*Rag2*$^{-/-}$ or OTII-*Rag2*$^{-/-}$x*Ltα*$^{-/-}$ recipients. Clonal deletion was analysed 3–5 days after AT. **e** Numbers of total thymic cells, DP and Vβ5$^+$Vα2$^+$CD4$^+$ SP cells were analysed for each condition. *UT* untreated OTII-*Rag2*$^{-/-}$ mice. Data are representative of three independent experiments ($n = 4$ mice per group and per experiment). *MΦ* macrophage. Error bars show mean ± SEM, *$p < 0.05$, **$p < 0.01$, ***$p < 0.001$, ****$p < 0.0001$ using unpaired Student's *t*-test for **b**, two-tailed Mann–Whitney test for **c** and one-tailed Mann–Whitney test for **e**.

tuning mechanism that allows the thymus to adapt its capacity of deleting autoreactive T cells to physiological and pathological fluctuations. This study should open new therapeutic perspectives for autoimmune disorders, based on the deletion of hazardous T cells via the manipulation of thymic entry of peripheral DCs and macrophages.

## Methods

**Mice.** CD45.1 WT (B6.SJL-*Ptprc*[a] *Pepc*[b]/BoyCrl, Stock n°002014, Charles River), CD45.2 WT (Stock n°000664, Charles River), CD45.2 *Ltα*[−/−][58], CD45.2 *Ccr2*-red fluorescent protein (RFP)/RFP (CCR2[RFP/RFP])[59], CD45.2 *Ccr1*[−/−][60], CD45.2 *Ccr5*[−/−][61], OTII[62] and RipmOVA[27] mice were on a C57BL/6J background. OTII and RipmOVA mice were backcrossed on a *Rag2*[−/−] background and OTII-*Rag2*[−/−] mice on a *Ltα*[−/−] background (OTII-*Rag2*[−/−]x*Ltα*[−/−] mice). All mice were maintained under specific pathogen-free condition at the Centre d'Immunologie de Marseille-Luminy, France. Standard food and water were given *ad libitum*. For all experiments, males and females were used at the age of 6 weeks. Mice were killed using CO$_2$ and terminated via cervical dislocation. All experiments were done in accordance with national and European laws for laboratory animal welfare (EEC Council Directive 2010/63/UE), and were approved by the Marseille Ethical Committee for Animal Experimentation (Comité National de Réflexion Ethique sur l'Expérimentation Animale no. 14).

**Bone marrow chimeras.** After flushing with RPMI medium, $5 \times 10^6$ BM cells from tibia and femurs of OTII-*Rag2*[−/−] mice were injected *i.v.* into lethally γ-irradiated OTII-*Rag2*[−/−] or RipmOVA-*Rag2*[−/−] recipients (two doses of 500 rads, 5 h apart; X-ray; RS-2000 Irradiator; Rad Source Technologies). The same experiment was performed by transplanting $5 \times 10^6$ BM cells from CD45.2 WT or *Ltα*[−/−] mice into lethally γ-irradiated CD45.1 WT recipients. For mixed bone marrow chimeras, $3 \times 10^6$ BM cells from either CD45.1 WT mice were co-injected *i.v.* with $3 \times 10^6$ BM cells from CD45.2 WT, *Ccr2*[RFP/RFP], *Ccr1*[−/−] or *Ccr5*[−/−] mice into lethally γ-irradiated CD45.1xCD45.2 WT recipients. Chimeras were generated at 6–8 weeks of age. All BM transplants were performed with sex and age-matched mice.

**Cell isolation.** Thymus, spleen and lymph nodes were digested for 15 min at 37 °C in HBSS medium containing 1 mg/ml of collagenase D and 100 µg/ml of DNase I (Roche), and were subjected to vigorous pipetting. This step was repeated until complete tissue digestion. Cells were filtered through a 70 µm mesh to remove clumps. Splenic red blood cells were lysed with RBC lysis buffer (eBioscience). Total mTECs were purified by sorting CD45−Ep-CAM+BP-1[lo]UEA-1+ cells after depletion of CD45 haematopoietic cells, using anti-CD45 magnetic beads by autoMACS with the DepleteS program (Miltenyi Biotec). mTEC[lo] and mTEC[hi] cells were sorted as CD80[lo] and CD80[hi] mTECs, respectively. Flow cytometry gating strategies are shown in Fig. 5a. For co-culture experiments, resident cDCs, migratory cDCs and pDCs were purified from the thymus of WT mice, by sorting SHPS-1−CD11c[hi]BST-2[lo], SHPS-1+CD11c[hi]BST-2[lo] and CD11c[int]BST-2[hi], respectively. Flow cytometry gating strategies are shown in Supplementary Fig. 1. CD4+ thymocytes from OTII or OTIIx*Ltα*[−/−] mice were purified by sorting CD4+CD8− cells after depletion of CD8+ cells, using biotinylated anti-CD8 antibody with anti-biotin microbeads by AutoMACS via the DepleteS program (Miltenyi Biotec). To analyse their thymic distribution after AT, cDCs, pDCs and macrophages were purified from the spleen and lymph nodes of *Ccr2*[RFP/+] mice by sorting of SHPS-1+CD11c[hi]BST-2[lo], CD11c[int]BST-2[hi] and F4/80+CD11b+ cells, respectively. Flow cytometry gating strategies are shown in Supplementary Fig. 14a,b. BM-derived cDCs, pDCs and macrophages, defined as CD11c[hi]MHCII+, CD11c+BST-2[hi] and F4/80+CD11b+, respectively were cell-sorted. Flow cytometry gating strategies are shown in Supplementary Fig. 14c-e. All purified cell types were cell-sorted (purity >95%) using a FACSAria III cell sorter (BD).

**Flow cytometry.** DCs, macrophages, mTECs and thymocytes were analysed by flow cytometry with standard procedures. Cells were incubated for 15 min at 4 °C with Fc-block (anti-CD16/CD32, 2.4G2, BD Biosciences) before staining with the following antibodies: anti-CD11c (N418), anti-SHPS-1 (anti-Sirpα, P84), anti-BST-2 (anti-PDCA-1, 29 F.1A12), anti-CD8α (53-6.7), anti-MHCII (I-Ab, AF6-1201), anti-CD4 (RM4-5), anti-F4/80 (6F12), anti-TCR Vα2 (B20.1), anti-Helios (22F6), anti-CCR7 (4B12), anti-CD69 (H1.2F3), anti-PD-1 (29 F.1A12 FC), anti-CD45.2 (104), anti-CCR1 (S15040E), anti-CCR5 (HM-CCR5), anti-NF-κB p65 (D14E12) and anti-phospho NF-κB p65 (Ser536; 93H1) were purchased from BioLegend. Anti-RelB (C-19) and anti-MCP1–4 (B-2) antibodies were purchased from Santa Cruz. Anti-CD62L (MEL-14), anti-CD80 (16-10A1), anti-BP-1 (anti-Ly51, BP-1), anti-CD45.1 (A20), anti-CD11b (M1/70), anti-CD45 (30-F11) and anti-TCR Vβ5.1,5.2 (MR9-4) antibodies were purchased from BD Biosciences. Anti-CCR2 (475301) antibody was purchased from RnD systems. FITC-conjugated UEA-1 lectin was purchased from Vector Laboratories. Anti-Ep-CAM (G8.8), anti-Foxp3 (FJK-16s), anti-Ki-67 (SolA15) and anti-LTβR (ebio3C8) antibodies were purchased from eBioscience. For intracellular staining of anti-Foxp3, anti-Helios, anti-Ki-67, anti-NF-κB p65, anti-phospho NF-κB p65, anti-RelB and anti-MCP1–4, cells were fixed, permeabilized and stained with the Foxp3 staining kit, according to the manufacturer's instructions (eBioscience). Cells

were incubated 1 h at 37 °C with anti-CCR1 and anti-CCR5 antibodies. For MCP1–4 detection, cells were incubated for 3 h with brefeldin A (BioLegend). Anti-MCP1–4 staining was revealed with Alexa Fluor 488-conjugated goat anti-mouse IgG and anti-NF-κB p65; anti-phospho NF-κB p65 and anti-RelB were revealed with Alexa Fluor 647-conjugated donkey anti-rabbit IgG (Invitrogen). All antibodies and reagents are described in Supplementary Table 2. Fluorescence minus one (FMO) and secondary antibody controls were used to set positive staining gates. Flow cytometry analysis was performed with a FACSCanto II (BD) and data were analysed using FlowJo software (BD).

**In vitro co-culture assays.** All $2 \times 10^3$ cell-sorted CD45−Ep-CAM+BP-1[lo]UEA-1+ mTECs, SHPS-1+CD11c[hi]BST-2[lo] cDCs, SHPS-1−CD11c[hi]BST-2[lo] cDCs and CD11c[int]BST-2[hi] pDCs from WT thymi were loaded or not for 1 h with OVA$_{323–339}$ peptide (10 µg per ml, Polypeptide group) at 37 °C. Cells were then cultured for 24 h with $1 \times 10^4$ purified CD4+ thymocytes, from OTII-*Rag2*[−/−] or OTII-*Rag2*[−/−]x*Ltα*[−/−] mice, in a medium containing RPMI (ThermoFisher) and 10% FBS (Sigma Aldrich) supplemented with L-glutamine (2 mM, ThermoFisher), sodium pyruvate (1 mM, ThermoFisher), 2-mercaptoethanol ($2 \times 10^{-5}$ M, ThermoFisher), penicillin (100 IU per ml, ThermoFisher) and streptomycin (100 µg per ml, ThermoFisher). When indicated, recombinant LTβR-Fc chimera (2 µg per ml, RnD systems) was added to the culture medium.

**Adoptive transfer of blood and splenic CD45.1 donor cells.** Red blood cells from CD45.1 WT congenic mice or *Ccr2*[RFP/RFP] homozygous mice were lysed with RBC lysis buffer (eBioscience). All $3 \times 10^6$ nucleated blood cells from CD45.1 WT congenic mice were injected *i.v.* into sublethally γ-irradiated OTII-*Rag2*[−/−], RipmOVAxOTII-*Rag2*[−/−], CD45.2 WT or *Ltα*[−/−] recipients (one dose of 500 rads; X-ray; RS-2000 Irradiator; Rad Source Technologies). For co-adoptive transfer experiments, $6 \times 10^6$ nucleated blood cells from CD45.1 WT congenic mice and *Ccr2*[RFP/RFP] homozygous mice were co-injected *i.v.* into sublethally γ-irradiated CD45.2 WT or *Ltα*[−/−] recipients (ratio 1:1; one dose of 500 rads; X-ray; RS-2000 Irradiator; Rad Source Technologies). For AT of DC and macrophage-enriched cells, spleens and lymph nodes from CD45.1 WT congenic mice or *Ccr2*[RFP/RFP] homozygous mice were depleted in T and B cells, using biotinylated anti-CD3 and anti-CD19 antibodies with anti-biotin microbeads by AutoMACS via the deplete program (Miltenyi Biotec). All $5–10 \times 10^6$ DC and macrophage-enriched cells were injected *i.v.* into unmanipulated OTII-*Rag2*[−/−], RipmOVAxOTII-*Rag2*[−/−], WT or *Ltα*[−/−] recipients. Sex and age-matched adult mice of 6 weeks were used for all adoptive transfer experiments.

**Bone marrow-derived cDCs/pDCs/macrophages.** After red blood cell lysis with RBC lysis buffer (eBioscience), BM cells from WT mice were cultured for 7–10 days in complete RPMI medium containing murine M-CSF (1 ng per ml, PeproTech), murine Flt3-ligand (100 ng per ml, PeproTech) and GM-CSF (20 ng per ml, PeproTech) for the generation of cDCs, pDCs and macrophages. Cellular differentiation was assessed by flow cytometry.

**Immunofluorescence staining.** Thymi were harvested and frozen in O.C.T. (Sakura Finetek). The 20-µm sections were fixed with 2% paraformaldehyde (Merck). Thymic slices were stained, as previously described, with the following antibodies: Alexa Fluor 488-conjugated anti-F4/80 (6F12; BD Biosciences), biotinylated anti-CD11c (N418; BioLegend), Alexa Fluor 488-conjugated anti-BST-2 (129c, eBioscience) and rabbit anti-keratin 14 (AF64; Covance Research). Biotinylated anti-CD11c and anti-K14 were revealed with Alexa Fluor 488-conjugated streptavidin and Cy5-conjugated anti-rabbit IgG or Alexa Fluor 488-conjugated anti-rabbit IgG (Invitrogen), respectively. Sections were counterstained with DAPI (BioLegend) and mounted with Mowiol (Calbiochem). Immunofluorescence confocal microscopy was performed with a LSM 780 Leica SP5X confocal microscope.

**Confocal image analysis.** Confocal microscopy images were segmented using homemade Matlab scripts. Twenty-five images for *Ltα*[−/−] thymic sections with AT of purified DCs and macrophages from *Ccr2*[RFP/+] mice and six to eleven images for WT and *Ltα*[−/−] thymic sections were analysed. Briefly, images were sequentially treated by: (i) median filtering over 2 pixels, (ii) thresholding, using the automatic threshold value (according to Otsu method, as implemented in the Matlab function graythresh), (iii) smoothing (dilatation followed by erosion) over 3 pixels, and (iv) background subtraction, with the background estimated by median filtering of the colocalization image over 32 pixels. For cells expressing RFP, colocalization between the green (endogenous DC or macrophage staining) and red channels (RFP) was estimated as the minimum intensity of the images of both channels. This ensured validating signals arising only from high staining for both colours before proceeding to cell detection. In all cases, cells were detected from the resulting binary images. The medulla was detected by the K14 staining by median filtering over 7 pixels, automatically thresholding, filling holes and smoothing over 30 pixels. Defining a binary mask corresponding to the medulla and to the cortex allowed automated counting of cells in each compartment. Cell density was computed as cell number over surface (d = N/S), for medulla and cortex, allowing computing the ratio of medulla versus cortex density (ratio = $d_{\mathrm{medulla}}/d_{\mathrm{cortex}}$).

**Quantitative RT-PCR**. Total RNA was prepared with TRIzol reagent (Invitrogen). cDNA was synthesised with oligo dT primers (Life Technologies) and Superscript II reverse transcriptase (Invitrogen). PCR was performed using SYBR Premix Ex Taq (Takara) with the ABI Prism 7500 Fast PCR System (Applied Biosystems). Actin mRNA was used for normalisation. Primers are listed in Supplementary Table 3.

**Analysis for NF-κB binding sites in the Ccl12 promoter**. In silico analysis of the mouse Ccl12 promoter was performed using the software MatInspector (https://www.genomatix.de). The promoter region from −1000 to +100 bp of the transcription start site was searched for NF-κB binding sites using the Matrix family library version 10.0.

**Statistics**. All data are presented as means ± standard error of mean (SEM). Statistical analysis was performed with GraphPad Prism 7.03 software using unpaired Student's $t$-test for normal distribution or Mann-Whitney test. ****$p < 0.0001$; ***$p < 0.001$; **$p < 0.01$; *$p < 0.05$. Normal distribution of the data was assessed using d'Agostino-Pearson omnibus normality test.

**Data availability**. The authors declare that the data supporting the findings of this study are available within the article and its supplementary information files, or are available upon reasonable requests to the authors.

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

## Acknowledgements

We thank Prof. Philippe Naquet and his team for discussions. We thank Drs. Stéphanie Hugues and Julien Marie for critical reading of the manuscript, Alexis Combes and Sylvie Memet for advices and the animal and flow cytometry facilities at the Centre d'Immunologie de Marseille-Luminy for technical support. We are grateful to Dr. Christophe Combadiere for providing us *Ccr1*⁻/⁻ and *Ccr5*⁻/⁻ BM cells. This work was supported by institutional grants from Institut National de la Santé et de la Recherche Médicale, Centre National de la Recherche Scientifique and Aix-Marseille Université, the Swiss National Science Foundation (PZ00P3-131945 to M.I.), the Marie Curie Actions (Career Integration Grants, CIG_SIGnEPI4Tol_618541 to M.I.), the Jules Thorn Foundation (to M.I.), Novonordisk Foundation (6131 to O.C.) and Augustinus Foundation (14-2913 to O.C.). We acknowledge financial support from n°ANR-10-INBS-04-01 France Bio Imaging.

N.L. and J.C. are supported by a PhD fellowship from Aix-Marseille University and the Ministère de l'Enseignement Supérieur et de la Recherche, respectively.

## Author contributions

N.L., J.C., O.C. and M.I. performed experiments and analysed the data. A.S. analysed the data. N.L. and M.I. interpreted data and wrote the manuscript. M.I. initiated, supervised and conceived the study.

## Additional information

**Competing interests:** The authors declare no competing interests.

