## [Peer Review File(PDF 915 kb) · Nature Communications]

Reviewers' comments:

Reviewer #1 (Thymic selection, TEC)(Remarks to the Author):

This manuscript tries to describe that thymocyte-mTEC interaction regulates the entry of DCs and macrophages into the thymus via regulating LT and CCR2 signals. The scope of the study can be important and the conclusions offered in the manuscript are potentially interesting. However, key results shown and major conclusions described in this manuscript are not compelling.

1) The results from OTII vs RIP-mOVA experiments cannot be readily interpreted as indicating that RIP-mOVA-expressing mTECs directly interact with OTII TCR transgenic thymocytes. DC-mediated indirect antigen presentation may be involved in the interaction between mTECs and thymocytes, at least in part (as suggested by ref 12 and 31). Possible involvement of DCs within the thymocyte-mTEC cross-talk would greatly alter major conclusions of the study (drawn in supplementary figure 5).

2) In addition to LT, RANKL and CD40L are well known to mediate the thymocyte-mTEC cross-talk (for example, ref 5). The manuscript offers no logic for the focus only on to LT in this study. How do RANKL and CD40L affect DC entry to the thymus? Are they important like LT?

3) The results equally show the elevation of CCL2, CCL8, and CCL12 in mTECs without LT signals (Figure 3D). What is the logic for the focus only on to CCR2 (CCL2/CCL12 receptor) in this study? What are the roles for CCR1 and CCR5 (CCL8 receptors)?

Figure 1C: Are you erasing the dots in CD11c low areas? You should not manipulate the data by arbitrarily erasing the data. You can adequately gate for CD11c-high events and show single histograms for Sirpa.

CD45.1 staining profiles shown in many data panels (e.g. Figure 1G, H, I, 2G, 3G, 3I) are poor without controls. In case CD45 expression levels are low in those cells, the single histograms for CD45.1 are not appropriate for distinguishing between the transferred cells. You may want to use simultaneous staining with CD45.1 and CD45.2 antibodies for better separation.

Reviewer #2 (Thymocyte, apoptosis)(Remarks to the Author):

This manuscript investigates the mechanism by which peripheral APCs enter the thymus to participate in clonal deletion, in particular the role of lymphotoxin alpha (LTa) in the process.

The authors successively show that CD4⁺/mTECs interactions regulate the homing of peripheral DCs and macrophages to the thymus; that LTa limits the accumulation of these cells in the thymus; that Ccl2, Ccl8 and Ccl12 are expressed more strongly in the absence of LTa; that loss of CCR2 abrogates homing of DCs and macrophages to the thymus; that increased numbers of DCs and macrophages increase clonal deletion.

The novelty obviously resides in the description of the role of LTa in limiting the recruitment of DCs and macrophages to the thymus and the proposed role of Ccl2, Ccl8 and Ccl12 downstream of LTa. How LTa increases the expression of these cytokines is not addressed.

I am somewhat surprised by the consistently small size of the error bars throughout the figures, and I don't really understand the meaning of 'data representative of x independent experiments (n = 3 mice per group)'. Does each independent experiment have 3 mice per group, or is n = 3 for the total of all independent experiments? Having worked with mice for a long time, I would have expected larger variations in at least some of the experiments, and I think that n = 3 is insufficient for that type of work.

I know that the OTII system is CD4-specific, but I believe the study would have more impact if the deletion of LTa was limited to CD4 cells.

The authors have a tendency to amplify the results. In the introduction, they talk about the 'massive' recruitment of DCs and macrophages into the thymus: most of the observed differences are in the 1.5 to 2-fold range, that is hardly 'massive'. For the same reason, the use of remarkably and strikingly may be restricted.

The OTI and OTII systems have been used extensively for the study of clonal deletion, and widely criticized because they do not recapitulate the diversity of the interactions antigen/TCR that exist in

non-transgenic mice. I certainly don't want to suggest that the authors repeat their experiments in another model. However, it would be interesting to know whether the loss of LTA would bring any benefit in a model of autoimmune disease due to impaired deletion. Would the increased numbers of DCs and macrophages somehow compensate and delay or abrogate the appearance of the symptoms? This may be part of the discussion.

In figure 2E-G and Supplementary figure 2: How do the numbers of DCs and macrophages that disappear from the blood of LTA^{-/-} compare with the increase of cell numbers found in the thymus? In other words, is the effect limited to the thymus or do some of these cells home to the spleen?

Authors' response

NCOMMS-17-09396

“mTEC-T cell crosstalk regulates via lymphotoxin α the thymic entry of peripheral APCs to fine-tune clonal deletion”

We warmly thank the Reviewers and Editors for their constructive comments. We believe that we have now extensively addressed the Reviewers' concerns, which have greatly improved our study. We provide below responses to all issues raised by the two Reviewers. The modifications in the text appear in red in the revised manuscript.

Reviewers' comments:

Reviewer #1 (Thymic selection, TEC)(Remarks to the Author):

This manuscript tries to describe that thymocyte-mTEC interaction regulates the entry of DCs and macrophages into the thymus via regulating LT and CCR2 signals. The scope of the study can be important and the conclusions offered in the manuscript are potentially interesting. However, key results shown and major conclusions described in this manuscript are not compelling.

1) The results from OTII vs RIP-mOVA experiments cannot be readily interpreted as indicating that RIP-mOVA-expressing mTECs directly interact with OTII TCR transgenic thymocytes. DC-mediated indirect antigen presentation may be involved in the interaction between mTECs and thymocytes, at least in part (as suggested by ref 12 and 31). Possible involvement of DCs within the thymocyte-mTEC cross-talk would greatly alter major conclusions of the study (drawn in supplementary figure 5).

Response:

We totally agree with the Reviewer that DCs could be involved in the interaction between mTECs and thymocytes through their ability to cross-present self-antigens expressed by mTECs and could thus be indirectly involved in the thymic entry of peripheral DCs and macrophages. In this revised version, we analyzed whether antigen-specific interactions between CD4⁺ thymocytes and mTECs alone, in absence of DCs, were sufficient to induce CCL2, CCL8 and CCL12 chemokines in mTECs. When we co-cultured OTII CD4⁺ thymocytes with OVA₃₂₃₋₃₃₉-loaded mTECs, we observed an increased expression of these three chemokines in mTECs, which was substantially enhanced by blocking LT α 1 β 2-LT β R interactions using a soluble LT β R-Fc fusion protein (cf. new Fig. 3e). Similar results were observed when mTECs were co-cultured with CD4⁺ thymocytes lacking LT α (CD4⁺ thymocytes from OTII-Rag2^{-/-} x LT α ^{-/-} mice; cf. new Fig. 3f). These new results indicate that direct antigen-specific interactions between mTECs and CD4⁺ thymocytes are sufficient to positively regulate the expression of CCL2, CCL8 and CCL12 and that the LT α 1 β 2-LT β R axis negatively controls the expression of these chemokines in mTECs.

Furthermore, in the regulation loop that we describe in this study (cf. new Supplementary Fig. 12), we have also investigated the possibility that DCs, *via* their antigen presentation capacity, could

induce LT α expression in CD4⁺ thymocytes and thus could be indirectly involved in the thymic entry of these peripheral APCs by negatively regulating CCL2, CCL8 and CCL12 chemokines in mTECs. To directly analyze the respective potential of the different thymic DC subsets in the induction of LT α expression, purified Sirp α ⁺ cDC, Sirp α ⁻ cDC and pDC loaded or not with OVA₃₂₃₋₃₃₉ peptide were co-cultured *in vitro* with OTII CD4⁺ thymocytes. In contrast to mTECs, these three DC subsets were inefficient in upregulating LT α in OTII CD4⁺ thymocytes, excluding thus a potential role of DCs in the regulation of these chemokines through LT α induction. This new result is shown in Fig. 3g.

In sum, these new data indicate that direct antigen-specific interactions between mTECs and CD4⁺ thymocytes are sufficient to regulate CCL2, CCL8 and CCL12 chemokine expression, which is fine tuned by the LT α 1 β 2-LT β R axis. Furthermore, we also provide evidence that the different DC subsets are inefficient in controlling LT α in CD4⁺ thymocytes and thus are unlikely involved in the thymic recruitment of peripheral DCs and macrophages.

2) In addition to LT, RANKL and CD40L are well known to mediate the thymocyte-mTEC cross-talk (for example, ref 5). The manuscript offers no logic for the focus only on to LT in this study. How do RANKL and CD40L affect DC entry to the thymus? Are they important like LT?

Response:

We apologize that our interest to focus this study on LT α was not sufficiently clear in the initial version of the manuscript.

Using OTII and RipmOVAxOTII transgenic mice in which OTII CD4⁺ thymocytes can interact or not in an antigen-specific manner with OVA-expressing mTECs (Kurts, C. *et al. J Exp Med* 184, 923-930 (1996), Anderson MS *et al., Immunity*. 2005 Aug;23(2):227-39), we found that mTEC-CD4⁺ T cell crosstalk controls numbers of Sirp α ⁺ cDCs, pDCs and F4/80⁺CD11b⁺ macrophages in the thymus by regulating their recruitment from blood (cf. new Fig. 1 and Supplementary Fig. 1 and 2).

We fully agree that the three TNF members, LT α , RANKL and CD40L constitute important signals implicated in crosstalk between mTECs and developing T cells. In this revised version, we analyzed whether the expression of these three TNF members are regulated upon mTEC-CD4⁺ T cell interactions. We found that in contrast to RANKL and CD40L, only LT α was upregulated in OTII CD4⁺ thymocytes upon antigen-specific interactions with mTECs. These new results are now shown in Fig. 2a and are fully consistent with our previous study (Irla *et al. PLoS One*. 2012;7(12):e52591), indicating that LT α is upregulated in CD4⁺ OTII thymocytes from RipmOVAxOTII mice. These data thus indicate that among these three TNF members, only LT α is regulated by crosstalk with mTECs, suggesting that LT α could be a good candidate involved in the regulation of thymic entry of peripheral DCs and macrophages, mediated by mTEC-T cell crosstalk. By adding these new data (cf. Fig. 2a), we thus now clarified our interest to specifically focus this study on LT α in the context of mTEC-T cell crosstalk. Nevertheless, we also commented in the discussion part that a potential role of RANKL and CD40L in other aspects of DC biology cannot be excluded at this stage (cf. page 14) and that future investigations are required to clarify this issue.

3) The results equally show the elevation of CCL2, CCL8, and CCL12 in mTECs without LT signals

(Figure 3D). What is the logic for the focus only on to CCR2 (CCL2/CCL12 receptor) in this study? What are the roles for CCR1 and CCR5 (CCL8 receptors)?

Response:

We thank the Reviewer for this suggestion. As requested, in addition to CCR2, we investigated in this revised version the potential implication of CCR1 and CCR5 in regulating numbers of Sirpα⁺ cDCs, pDCs and F4/80⁺CD11b⁺ macrophages into the thymus. We first found by flow cytometry that these three cell-types express these two chemokine receptors in the blood from both WT and LTα^{-/-} mice (cf. new Supplementary Fig. 7). To analyze the role of CCR1 and CCR5 in controlling the thymic pool of these cells, we generated mixed bone marrow chimeras in which lethally irradiated CD45.1xCD45.2 WT recipient mice were reconstituted with equal numbers of BM cells from CD45.1 WT mice and from either CD45.2 CCR1^{-/-}, CCR5^{-/-} or CCR2^{RFP/RFP}-deficient mice (cf. new Fig 4a). This experiment revealed that CCR5 plays a non-negligible role in the regulation of Sirpα⁺ cDCs, pDCs and macrophages in the thymus while CCR1 seems to play a minor role in Sirpα⁺ cDC and pDC recruitment (cf. new Fig 4b). We found a ~1.5-fold reduction for Sirpα⁺ cDCs and pDCs in the CCR1^{-/-} donor group whereas a ~1.5 to 2-fold reduction of these three cell types was observed in the CCR5^{-/-} donor group compared to the CD45.2 WT control group. Nevertheless, the effects of CCR1 and CCR5 deficiencies were less pronounced than that of CCR2. A ~6, 5 and 12-fold reduction in Sirpα⁺ cDCs, pDCs and macrophages, respectively were observed in the CCR2^{RFP/RFP}-deficient group (cf. new Fig. 4b), indicating that CCR2 plays a major role in regulating the thymic cellularity of these three cell types. To our knowledge, since no functional role for CCR1 and CCR5 in the regulation of these cell types in the thymus has been described to date, we thus provide the first evidence that in addition to CCR2, CCR1 and CCR5 are partially involved in controlling the thymic pool of Sirpα⁺ cDCs, pDCs and macrophages. Since we found that CCR2 plays a major role in regulating the cellularity of these three cell types, these observations motivated further experiments based on the adoptive transfer of nucleated blood donor cells derived from CCR2^{RFP/RFP}-deficient mice in LTα^{-/-} mice in order to analyze whether increased migration of these three cell-types depends on this chemokine receptor (cf. new Fig. 4c-e). We found that the increased migration of cDCs, pDCs and macrophages observed in LTα^{-/-} mice was drastically impaired with donor cells lacking CCR2, thus showing that LTα regulates the thymus homing of these peripheral cells mainly in a CCR2-dependent manner.

Figure 1C: Are you erasing the dots in CD11c low areas? You should not manipulate the data by arbitrarily erasing the data. You can adequately gate for CD11c-high events and show single histograms for Sirpa.

Response:

We apologize if our gating strategy was not sufficiently clear in the initial version of our manuscript. In the original Fig. 1c (corresponding to the new Supplementary Fig. 1c of this revised version), we have not erased the dots in CD11c low areas. We have only gated on CD11c^{hi} cells (as described in the original Fig. 1b or the new Supplementary Fig. 1b) to further analyze the conventional DC population based on Sirpα expression. As requested by the Reviewer, you will find below a detailed explanation of our gating strategy.

As depicted on this panel, gating on CD11c^{hi} cDCs allows the identification of Sirpα⁺ cDCs represented here either as a CD11c^{hi}/Sirpα⁺ dot-plot or as a single histogram for Sirpα staining, as requested by the Reviewer. In contrast, gating on CD11c⁺ total cells shows that CD11c^{int} cDCs do not contain Sirpα⁺ cDCs as represented both by a CD11c^{hi}/Sirpα⁺ dot plot and a single histogram for Sirpα. Classically, cDCs in the thymus are defined as CD11c^{hi} cells (cf. Proietto *et al.* Proc Natl Acad Sci U S A. 2008 Dec 16;105(50):19869-74). Thymic cDCs were therefore defined as CD11c^{hi} cells in our study.

CD45.1 staining profiles shown in many data panels (e.g. Figure 1G, H, I, 2G, 3G, 3I) are poor without controls. In case CD45 expression levels are low in those cells, the single histograms for CD45.1 are not appropriate for distinguishing between the transferred cells. You may want to use simultaneous staining with CD45.1 and CD45.2 antibodies for better separation.

Response:

As suggested by the Reviewer, these experiments of adoptive transfer with donor cells were reproduced with a simultaneous staining using CD45.1 and CD45.2 antibodies (cf. new Fig. 1f, Fig. 2f, Supplementary Fig. 2, Supplementary Fig. 5 and Supplementary Fig. 9a, the latter related to the new Fig. 4d). As depicted in these new panels, the Reviewer was right that donor cells express different levels of CD45 expression. By using a double staining for CD45.1 and CD45.2 markers, we were able to unambiguously identify CD45.1 donor cells in this revised version. Among CD45.1^{lo/hi}CD45.2⁻ donor cells, we have analyzed the presence of cDCs, pDCs and macrophages (cf. new Fig. 1g,h; Fig. 2g,h, Fig.

4e, Supplementary Fig 2 and 5). We thank the Reviewer for pointing that out. Furthermore, as recommended by the Reviewer, we have now included control mice (*i.e.* non-injected OTII mice in Fig. 1f and non-injected WT mice in Fig. 2f). These experiments of adoptive transfer in OTII and RipmOVAxOTII mice as well as in WT and $LT\alpha^{-/-}$ mice were not only performed with splenic DC and macrophages-enriched cells (*cf.* new Supplementary Fig. 2 and 5, respectively) but also with nucleated blood cells from CD45.1 congenic mice (*cf.* Fig. 1f-h and Fig. 2f-h), the latter experiment being certainly more relevant to the migration process observed under physiological conditions from blood to the thymus (as performed in WT recipients in the study of Li J *et al.* *J Exp Med.* 2009 Mar 16;206(3):607-22). The increased migration of peripheral DCs and macrophages in the thymus of RipmOVAxOTII and $LT\alpha^{-/-}$ mice compared to OTII and WT mice, respectively were observed with both splenic DC and macrophages-enriched cells and nucleated blood cells, thus reinforcing our conclusions.

Reviewer #2 (Thymocyte, apoptosis)(Remarks to the Author):

This manuscript investigates the mechanism by which peripheral APCs enter the thymus to participate in clonal deletion, in particular the role of lymphotoxin alpha (LTa) in the process. The authors successively show that CD4+/mTECs interactions regulate the homing of peripheral DCs and macrophages to the thymus; that LTa limits the accumulation of these cells in the thymus; that Ccl2, Ccl8 and Ccl12 are expressed more strongly in the absence of LTa; that loss of CCR2 abrogates homing of DCs and macrophages to the thymus; that increased numbers of DCs and macrophages increase clonal deletion.

The novelty obviously resides in the description of the role of LTa in limiting the recruitment of DCs and macrophages to the thymus and the proposed role of Ccl2, Ccl8 and Ccl12 downstream of LTa. How LTa increases the expression of these cytokines is not addressed.

Response:

In this revised version, we attempted to investigate how the loss of LTa increases the expression of Ccl2, Ccl8 and Ccl12 chemokines in mTECs. First, since mTECs (defined as EpCAM⁺UEA-1⁺Ly51^{lo} cells) constitute an heterogenous population, we have examined which mTEC subset upregulates these three chemokines in $LT\alpha^{-/-}$ mice. In contrast to CD80^{hi} mTECs (mTEC^{hi}), we found that only CD80^{lo} mTECs (mTEC^{lo}) upregulated these chemokines in absence of LTa (*cf.* new Fig. 5b). We also confirmed by flow cytometry using an anti-MCP1-4 antibody (which recognizes CCL2, CCL7, CCL8 and CCL13, the latter being not expressed in mice) that this cell type expressed higher levels of MCP1-4 proteins in $LT\alpha^{-/-}$ mice than in WT mice (*cf.* new Fig. 5c). Previous studies have shown that CCL2 and CCL8 expression is regulated by the classical NF-KB pathway (Ping, D., *et al.* *Immunity* 4, 455-469 (1996), Ueda, A. *et al.* *J. Immunol.* 153, 2052-2063 (1994), and Ueda, A. *et al.* *The Journal of biological chemistry* 272, 31092-31099 (1997)). Although the transcription factors that regulate Ccl12 expression remain enigmatic to date, we identified in this revised version two putative NF-KB binding sites for c-Rel and p65 in the promoter region of the Ccl12 gene (*cf.* new Supplementary Table 1), suggesting that this gene could also be regulated by the classical NF-KB pathway. Moreover, $LT\alpha 1\beta 2/LT\beta R$ signaling axis is known to activate the non-classical NF-KB pathway (Gommerman, J. L.

& Browning, J. L. *Nat Rev Immunol* 3, 642-655 (2003) and Irla, M., *Trends in immunology* 31, 71-79, (2010)). We therefore analyzed whether mTEC^{lo} cells from LT α ^{-/-} mice show a dysregulation in the expression of classical and non-classical NF- κ B subunits that could explain CCL2, CCL8 and CCL12 upregulation. We found both at mRNA and protein levels that while the non-classical NF- κ B subunit Relb was decreased, the classical NF- κ B subunits cRel and p65 were substantially enhanced in mTEC^{lo} cells from LT α ^{-/-} mice compared to WT mice (cf. new Fig. 5e-g). These results thus indicate that the upregulation of CCL2, CCL8 and CCL12 in mTEC^{lo} from LT α ^{-/-} mice correlates with the upregulation of classical NF- κ B subunits.

Furthermore, given that co-culture experiments using OVA₃₂₃₋₃₃₉-loaded mTECs with CD4⁺ thymocytes from OTI \times LT α ^{-/-} mice resulted in the upregulation of these chemokines (cf. new Fig. 3f), we also analyzed the effect of LT α 1 β 2/LT β R axis upon crosstalk with CD4⁺ T cells in the regulation of NF- κ B subunits in mTEC^{lo}. Consistently with our results based on purified mTEC^{lo} from LT α ^{-/-} mice, we found a downregulation of RelB and an upregulation of cRel and p65 when mTECs were co-cultured with OTI \times LT α ^{-/-} CD4⁺ thymocytes (cf. new Fig. 5i). Altogether, these results show a concomitant upregulation of CCL2, CCL8 and CCL12 as well as cRel and p65 classical NF- κ B subunits, suggesting that the absence of LT α leads to the upregulation of these chemokines in mTEC^{lo} cells through the classical NF- κ B pathway usage, previously described to regulate these chemokines at the transcriptional level. Nevertheless, to definitively prove a direct regulation of these chemokines by classical NF- κ B members, experiments based on chromatin immunoprecipitation would be required. Such experiments are technically challenging to date since the mTEC^{lo} subset constitutes a rare cell type in the thymus ($\sim 2 \times 10^5$ cells/thymus). This point has been commented in the discussion part (cf. page 14).

I am somewhat surprised by the consistently small size of the error bars throughout the figures, and I don't really understand the meaning of 'data representative of x independent experiments (n = 3 mice per group)'. Does each independent experiment have 3 mice per group, or is n = 3 for the total of all independent experiments? Having worked with mice for a long time, I would have expected larger variations in at least some of the experiments, and I think that n = 3 is insufficient for that type of work.

Response:

We apologize for this confusion. Our sentence in figure legends 'data representative of x independent experiments (n = 3 mice per group)' means that we have performed x independent experiments with 3 mice per experiment and per group. For instance, "Data are representative of three independent experiments (n = 3 mice per group)" means that we have analyzed n = 9 mice in total. Consequently, for each experiments performed in this study, we have analyzed between 6 and 15 mice per group in total. For a better clarity, in this revised version, we have indicated in the figure legends "n = X mice per group and per experiment".

Furthermore, as indicated in the method section and figure legends, we have used standard error of mean (SEM), which is by definition smaller than the standard deviation (SD), given that SEM takes into account both the value of the SD and the sample size.

I know that the OTII system is CD4-specific, but I believe the study would have more impact if the deletion of LTa was limited to CD4 cells.

Response:

We understand the point raised by the Reviewer. Unfortunately, we do not have the possibility to analyze transgenic mice with a specific deletion for the LTa gene in CD4⁺ T cells. Nevertheless, to further analyze whether the specific loss of LTa in CD4⁺ thymocytes controls CCL2, CCL8 and CCL12 levels in mTECs, we performed *in vitro* co-culture with OVA₃₂₃₋₃₃₉-loaded mTECs and purified CD4⁺ thymocytes from either OTII or OTIIxLTa^{-/-} mice. We found that CCL2, CCL8 and CCL12 expression was enhanced when mTECs were co-cultured with LTa-deficient CD4⁺ thymocytes (OTIIxLTa^{-/-} CD4⁺ thymocytes) compared to OTII CD4⁺ thymocytes. This new result thus confirms that the specific expression of LTa in CD4⁺ T cells is sufficient to negatively control CCL2, CCL8 and CCL12 in mTECs (cf. new Fig. 3f).

The authors have a tendency to amplify the results. In the introduction, they talk about the 'massive' recruitment of DCs and macrophages into the thymus: most of the observed differences are in the 1.5 to 2-fold range, that is hardly 'massive'. For the same reason, the use of remarkably and strikingly may be restricted.

Response:

We agree with the Reviewer that the differences observed in numbers of cDCs, pDCs and macrophages in the thymus are in a range of ~1.5 to 2-fold between OTII and RipmOVAxOTII mice (cf. new Fig. 1a-c; Fig. 1f-h) and between WT and LTa^{-/-} mice (cf. Fig. 2b-d; Fig. 2f-h; Fig. 4d,e; Fig. 7b,d,e). We have thus accordingly modified the text.

The OTI and OTII systems have been used extensively for the study of clonal deletion, and widely criticized because they do not recapitulate the diversity of the interactions antigen/TCR that exist in non-transgenic mice. I certainly don't want to suggest that the authors repeat their experiments in another model. However, it would be interesting to know whether the loss of LTa would bring any benefit in a model of autoimmune disease due to impaired deletion. Would the increased numbers of DCs and macrophages somehow compensate and delay or abrogate the appearance of the symptoms? This may be part of the discussion.

Response:

We thank the Reviewer for this suggestion. It would be indeed interesting to determine in a therapeutic perspective whether the loss of LTa by increasing thymic entry of DCs and macrophages shows a benefit in an autoimmune disease model due to impaired negative selection. It would be in particular interesting to use an inducible transgenic mouse model allowing the deletion of the LTa gene at a specific time point *i.e.* before or after the development of autoimmune signs. Such transgenic animals backcrossed with mice known to develop autoimmune disorders associated with defective T-cell selection are expected to determine whether the loss of LTa can protect and treat from autoimmunity. Unfortunately, to our knowledge such transgenic mice do not exist and it thus

seemed difficult to us to test this hypothesis. However, as suggested by the Reviewer, we have commented this point in the discussion part (cf. page 16).

In figure 2E-G and Supplementary figure 2: How do the numbers of DCs and macrophages that disappear from the blood of $LT\alpha^{-/-}$ compare with the increase of cell numbers found in the thymus? In other words, is the effect limited to the thymus or do some of these cells home to the spleen?

Response:

Our results of adoptive transfer experiments revealed that peripheral DCs and macrophages migrate more efficiently into the thymus of $LT\alpha^{-/-}$ mice compared to WT mice (cf. new Fig. 2e-h), which explains the increased numbers of these cell types observed in this tissue (cf. new Fig. 2b-d). Consequently, numbers of DCs and macrophages are reduced in the blood of $LT\alpha^{-/-}$ mice (cf. new Supplementary Fig. 6a-c). Furthermore, as requested by the Reviewer, we have examined in this revised version numbers of cDCs, pDCs and macrophages in the spleen from $LT\alpha^{-/-}$ mice and observed a reduction in these three cell types compared to WT mice (cf. new Supplementary Fig. 6d-f). These observations are consistent with the study of Wu Q *et al.* (*J Exp Med.* 1999 Sep 6;190(5):629-38) that has reported that $LT\alpha^{-/-}$ mice show markedly reduced numbers of CD11c⁺ DCs in the spleen. Altogether, these results show that increased numbers of DCs and macrophages in $LT\alpha^{-/-}$ mice are limited to the thymus.

REVIEWERS' COMMENTS:

Reviewer #1 (Remarks to the Author):

I think the revised manuscript is still weak. The new coculture experiments are artefactual and do not clarify the issues raised in my first two comments. The problems in flow cytometry data presentation (eg figure 4e, 5a) and poor CD45.1 profiles (eg figure 1f, 2f) persist.

Reviewer #2 (Remarks to the Author):

The authors have made a substantial effort to address the reviewers' comments and I believed that the manuscript has been improved in the process. I am happy to recommend the publication of this work.

Authors' response

NCOMMS-17-09396B

We would like to thank the Editors and Reviewers for this second round of review. We provide below our response to the comments raised by Reviewer #1.

REVIEWERS' COMMENTS:

Reviewer #1 (Remarks to the Author):

I think the revised manuscript is still weak. The new coculture experiments are artefactual and do not clarify the issues raised in my first two comments. The problems in flow cytometry data presentation (cf. figure 4e, 5a) and poor CD45.1 profiles (cf. figure 1f, 2f) persist.

We thank the reviewer for her/his comments. The initial first issue raised by the Reviewer was that DC-mediated indirect antigen presentation may be involved in the interaction between mTECs and thymocytes and thus in the thymic entry of DCs and macrophages. In the revised version, we have shown that antigen-specific interactions between CD4⁺ thymocytes and mTECs alone, in absence of DCs, were sufficient to induce CCL2, CCL8 and CCL12 chemokines in mTECs in co-culture experiments (cf. Fig. 3e). Moreover, in contrast to the distinct thymic DC subsets, only mTECs were capable to upregulate LT α expression in CD4⁺ thymocytes (cf. Fig. 3g).

Although informative, we understand that co-culture experiments could be somehow artefactual. In this new revised version, we have investigated whether thymic DCs are capable to express these three chemokines when mTECs express the cognate self-antigen for developing T cells (*i.e.* in the context of mTEC-thymocyte crosstalk) and thus whether they could be indirectly involved in the recruitment of peripheral APCs. Using an anti-MCP1-4 antibody (which recognizes CCL2, CCL7, CCL8 and CCL13, the latter being not expressed in mice), we found by flow cytometry that in contrast to mTECs, all thymic DC subsets (CD8 α ^{hi}Sirp α ⁻ resident cDCs, CD8 α ^{lo}Sirp α ⁺ migratory cDCs and pDCs) did not express detectable levels of MCP1-4 proteins in both OTII-*Rag2*^{-/-} and RipmOVAxOTII-*Rag2*^{-/-} mice (cf. new Supplementary Fig. 9a). Similar results were also observed with thymic DC subsets from WT and *Lt α* ^{-/-} mice (cf. new Supplementary Fig. 9b). These new data based on *ex vivo* purified cells thus indicate that thymic DC subsets are probably not involved in the recruitment of APCs by the secretion of CCR2 ligands.

In sum, our results show that:

- (i) mTECs alone co-cultured with CD4⁺ thymocytes in absence of DCs are sufficient to induce CCL2, CCL8 and CCL12 chemokines in mTECs,
- (ii) in contrast to DCs, only mTECs are capable to upregulate *Lt α* expression in CD4⁺ thymocytes,
- (iii) unlike to mTECs, DCs do not to express CCR2 ligands in the context of mTEC-CD4⁺ thymocyte crosstalk or upon LT α -mediated regulation.

These data suggest that DCs are unlikely able to attract peripheral APCs through the expression of CCR2 ligands or indirectly by regulating *Ltα* expression in CD4⁺ thymocytes.

Nevertheless, we cannot definitively rule out at this stage a role of DC-mediated indirect antigen presentation in the recruitment of peripheral DCs and macrophages into the thymus by other mechanisms. Future experiments based for example on the adoptive transfer of congenic donor cells in DC-depleted recipient mice in which mTECs express or not the cognate self-antigen for developing T cells are expected to answer this issue. This issue, which is not the subject of the present study is now mentioned in the “discussion” part (cf. page 15).

The second initial issue raised by the Reviewer was that in addition to LTα, RANKL and CD40L could affect DC entry into the thymus since these two TNF members are known to be also involved in the crosstalk between thymocytes and mTECs. Because we found that the thymic entry of peripheral DCs and macrophages is increased when mTECs express the cognate antigen for developing T cells (cf. Fig. 1), we have investigated the hypothesis that LTα, RANKL and CD40L could be differentially expressed in thymocytes upon crosstalk with mTECs and thus implicated in the thymus homing of these cell types. Co-culture experiments presented in the first revised version indicated that only *Ltα* was upregulated in CD4⁺ thymocytes upon antigen-specific interactions with mTECs (cf. Fig. 2a). We do not think that these results are artefactual since as mentioned in the text (cf. page 13), they are fully consistent with previously published observations from our group based on *ex vivo* purified DP and CD4⁺ thymocytes from OTII-*Rag2*^{-/-} and RipmOVAxOTII-*Rag2*^{-/-} mice (Irla *et al.* *PLoS One.* 2012;7(12):e52591; cf. Figure below).

Figure 6A from Irla *et al.* *PLoS One.* 2012;7(12):e52591.

Legend: RANKL, CD40L, LTα and LTβ mRNAs were quantified by qPCR in cell-sorted DP and CD4⁺ thymocytes from OTII-*Rag2*^{-/-} and RipmOVAxOTII-*Rag2*^{-/-} mice.

As depicted in the above figure, RANKL and CD40L were similarly upregulated from the double-positive (DP) to the CD4⁺ T cell stage in both transgenic mice and thus independently of OVA expression. In contrast to RANKL and CD40L, LTα was only upregulated in CD4⁺ thymocytes from RipmOVAxOTII-*Rag2*^{-/-} mice and not from OTII-*Rag2*^{-/-} mice, indicating that its induction relies on OVA-expressing mTECs.

Hence, since RANKL and CD40L are expressed to the same extent in CD4⁺ thymocytes in the presence or absence of OVA expression by mTECs (*i.e.* in RipmOVAxOTII-*Rag2*^{-/-} or OTII-*Rag2*^{-/-} mice, respectively), these two TNF members are unlikely responsible for the increased thymic entry of peripheral DCs and macrophages observed in RipmOVAxOTII-*Rag2*^{-/-} mice compared to OTII-*Rag2*^{-/-} mice (Cf. Fig. 1). In this revised version, we have clarified this point that justifies our interest to focus this study on LTα in the context of mTEC-T cell interactions since it is specifically induced upon crosstalk (cf. page 13). Nevertheless, we cannot rule out that RANKL and CD40L could be implicated in other aspects of thymic DC biology as discussed in the text (cf. page 13).

Figure 4e:

In this experiment of co-adoptive transfer of donor cells from CD45.1 WT mice and *Ccr2*^{RFP/RFP}-deficient mice into WT and *Lta*^{-/-} recipients (depicted in Fig. 4c), we have now included control mice *i.e.* sublethally non-injected WT mice to unambiguously identified donor cells by flow cytometry (cf. Fig. 4d, left panel). Moreover, this experiment (Fig. 4c-e) was repeated with two different FACS cell analysers: a FACSCanto II and a BD LSR II using different fluorochromes with distinct spectra. Dot-plots shown in the manuscript in Figure 4e were obtained with a FACSCanto II cell analyser, as indicated in the “Methods” section. In both cases, similar flow cytometry profiles were obtained, confirming the robustness of our results.

Figure 5a:

In this revised version, we provide the entire gating strategy that allowed us to identify mTEC^{lo} (CD80^{lo}) and mTEC^{hi} (CD80^{hi}) in WT and *Lta*^{-/-} mice. Total TECs were identified as Ep-CAM⁺ cells among CD45-negative enriched cells purified by autoMACS, as described in the “Methods” section. mTECs were then identified as UEA-1⁺Ly51^{lo} cells, as previously described by several groups (such as Uddin MM *et al.*, *Nat Commun.* 2017 Feb 8;8:14419, Jain R *et al.*, *Blood.* 2017 Dec 7;130(23):2504-2515, Žuklys S *et al.*, *Nat Immunol.* 2016 Oct;17(10):1206-1215, Hauri-Hohl M *et al.*, *Nat Immunol.* 2014 Jun;15(6):554-61, Wong *et al. Cell Rep.* 2014 Aug 21;8(4):1198-209 or Goldberg GL *et al. J Immunol.* 2010). mTEC^{lo} and mTEC^{hi} were then identified on the basis of CD80 expression level. Furthermore, we have adjusted the positioning of mTEC^{lo} and mTEC^{hi} gates in Fig. 5a.

Concerning the CD45.1 profiles depicted in Figures 1f and 2f:

We thank the Reviewer for pointing this concern out. As indicated in the point-by-point response of the first version of the manuscript, blood donor cells express different levels of the CD45.1 congenic marker. In these two experiments of adoptive transfer (Fig. 1f and Fig. 2f), the majority of donor cells expresses intermediate/high levels of the CD45.1 marker and thus shows a CD45.1^{int/hi} CD45.2⁻ phenotype. These observations on the CD45.1 expression level are fully consistent with the fact that blood pDCs from CD45.1 congenic mice express lower levels of the CD45.1 marker than cDCs and macrophages (cf. Figure below, which corresponds to the new Supplementary Fig. 3).

Importantly, this cell population is absent in the control condition (*i.e.* sublethally irradiated and non injected OTII-*Rag2*^{-/-} and WT mice, respectively), which attests their donor origin. Furthermore, we

would like to clarify that CD45.1^{int/lo}CD45.2⁻ cells correspond to endogenous stromal cells (Ep-CAM⁺ TECs, CD31⁺ endothelial cells and gp38⁺ fibroblasts) of recipient mice as depicted in the figure below:

In accordance with the control condition, we have thus adjusted more stringently the gates that identify CD45.1^{int/hi}CD45.2⁻ donor cells. Consequently, both experiments of adoptive transfer were re-analysed (cf. Fig 1f-h; Fig 2f-h). Importantly, these new analyses did not change the main conclusion of the previous version.

Reviewer #2 (Remarks to the Author):

The authors have made a substantial effort to address the reviewers' comments and I believed that the manuscript has been improved in the process. I am happy to recommend the publication of this work.

We thank Reviewer #2 for her/his favourable advice on our manuscript.